# Bayesian Inference-Based Estimation of Hourly Primary and Secondary Organic Carbon at Suburban Hong Kong: Multi-temporal Scale Variations and Evolution Characteristics during PM$_{2.5}$ episodes

Shan Wang[1#], Kezheng Liao[2#], Zijing Zhang[1], Yuk Ying Cheng[2], Qiongqiong Wang[3, 2], Hanzhe Chen[1], and Jian Zhen Yu[1,2*]

[1]Division of Environment and Sustainability, The Hong Kong University of Science and Technology, Clear Water Bay, Hong Kong, China

[2]Department of Chemistry, The Hong Kong University of Science and Technology, Clear Water Bay, Hong Kong, China

[3]School of Environmental Studies, China University of Geosciences, Wuhan, China

[#]These authors contributed equally to this work.

*Corresponding to*: Jian Zhen Yu (jian.yu@ust.hk)

**Abstract.** Observation-based data of primary and secondary organic carbon in ambient particulate matter (PM) are essential for model evaluation, climate and air quality research, health effects assessment, and mitigation policy development. Since there are no direct measurement tools available to quantify primary organic (POC) and secondary organic carbon (SOC) as separate quantities, their estimation relies on inference approaches using relevant measurable PM constituents. In this study, we measured hourly carbonaceous components and major ions in PM$_{2.5}$ for a year and a half in suburban Hong Kong from July 2020 to December 2021. We differentiated POC and SOC using a novel Bayesian inference approach. The hourly POC and SOC data allowed us to examine temporal characteristics varying from diurnal and weekly patterns to seasonal variations, as well as their evolution characteristics during individual PM$_{2.5}$ episodes. A total of 65 city-wide PM$_{2.5}$ episodes were identified throughout the entire study period, with SOC contributions during individual episodes varying from 10% to 66%. In summertime typhoon episodes, elevated SOC levels were observed during daytime hours, and high temperature and NO$_x$ levels were identified as significant factors contributing to episodic SOC formation. Winter haze episodes exhibited high SOC levels, likely due to persistent influences from regional transport originating from the northern region to the sampling site. Enhanced SOC formation was observed with increase in nocturnal NO$_3$ radical (indicated by the surrogate quantity of [NO$_2$][O$_3$]) and under conditions characterized by high water content and strong acidity. These results suggest that both NO$_3$ chemistry and acid-catalyzed aqueous-phase reactions likely make notable contributions to SOC formation during winter haze episodes. The methodology employed in this study for estimating POC and SOC provides practical guidance for other locations with similar monitoring capabilities in place. The availability of hourly POC and SOC data is invaluable for evaluating and improving atmospheric models, as well as understanding the evolution processes of PM pollution episodes. This, in turn, leads to more accurate model predictions and a better understanding of the contributing sources and processes.

## 1 Introduction

Carbonaceous aerosol is a major component of PM$_{2.5}$ (particulate matter with an aerodynamic diameter of less than 2.5 μm), accounting for 20-90% of its total mass in ambient environment (Seinfeld and Pandis, 1998; Kroll et al., 2011). It has been known to have adverse effects on regional to global climate, air quality, and human health (Nel, 2005; Bond et al., 2013; Huang et al., 2014a). Carbonaceous components can generally be classified into elemental carbon (EC) and organic carbon (OC). EC refers to the soot-like amorphous carbon emitted directly from incomplete combustion processes (Chow et al., 2010), while OC is a more complex mixture of organic compounds which can be either primarily emitted from anthropogenic sources (e.g., biomass burning, fossil fuel combustion, and cooking) and biogenic sources (e.g., plant debris), or secondarily formed through oxidation reactions (Donahue et al., 2009; Hallquist et al., 2009; Zhao et al., 2007; Christian et al., 2003). Therefore, OC can be further group to primary OC (POC) and secondary OC (SOC). Accurately quantifying and estimating POC and SOC through observation-based measurement is the precondition for comprehending their unique characteristics, such as relative contributions, temporal variations, and chemical evolution. This knowledge is crucial for refining atmospheric models and developing more targeted strategies to reduce carbonaceous aerosol emissions, mitigate climate change, and minimize human exposure.

The thermal-optical protocols have been widely used for OC and EC measurements (Klingshirn et al., 2019; Chow et al., 2001). However, accurately determining POC and SOC remains challenging since there are no instrument tools available for the direct measurement of POC and SOC. Several data treatment methodologies have been developed to estimate POC and SOC levels. One such approach is the chemical mass balance (CMB) receptor model, which apportions POC based on the chemical profile of individual known primary sources and the unmapped mass is then referred to as SOC (Pachon et al., 2010; Shi et al., 2011; Schauer and Cass, 2000). However, the uncertainty is large due to limited or insufficient information on the SOC source profiles in CMB simulations (Stone et al., 2009). Another widely used receptor model, positive matrix factorization (PMF), apportions the sources of OC based on the comprehensive chemical speciation data (Ke et al., 2008; Jaeckels et al., 2007). Studies have shown that PMF model output may underestimate the contributions of secondary organic aerosols when the specific molecular organic tracers are absent in the input data matrix (Wang et al., 2017; Pachon et al., 2010). These limitations compromise the applicability of receptor models in accurately quantifying the POC and SOC mass. Alternative approaches include the EC tracer method, which relies on the EC-to-OC ratio (Day et al., 2015; Turpin and Huntzicker, 1991), and the multiple linear regression (MLR) model (Blanchard et al., 2008). The former assumes that POC and EC share common combustion sources, allowing the POC/EC ratio to serve as an indicator to identify the primary sources, which can be determined by utilizing the minimum ratio (MIN) method (Castro et al., 1999). This assumption is less justified and compromised with naiveté, as observed OC/EC ratios can span over one order of magnitude in ambient measurements, which could be affected by measurement artifacts and fluctuate under different meteorological conditions (Yuan et al., 2006). Furthermore, the lack of a widely accepted criterion for percentile selection can bring up the bias to SOC estimation (Wu et al., 2019). The minimum R squared (MRS) method is a less arbitrary approach to determine the POC/EC ratio for primary sources. In MRS, the optimal primary POC/EC ratio is determined by minimizing the Pearson's correlation coefficient between EC and deduced SOC (Wu and Yu, 2016). The MRS method has been

increasingly used in studies with the hourly measurements in various environments (Wu et al., 2019; Yao et al., 2020; Bian et al., 2018). However, it has been proven that the MRS method inevitably yields a POC/EC ratio that renders EC and deduced SOC completely uncorrelated. This contradicts our expectation of a weak yet not negligible correlation between EC and SOC, since some SOC species could be formed from precursors co-emitted with EC through combustion activities (Jathar et al., 2013; Gentner et al., 2017; Deng et al., 2020), and both SOC and EC are influenced by regional transport or changes in the boundary layer height. On the other hand, the MLR model is a powerful statistical tool to estimate SOC by considering highly associated variables rather than the difference between measured OC and estimated POC (Kim et al., 2012; Pachon et al., 2010). Another approach, the secondary organic aerosol (SOA) tracer method, utilizes SOA tracer yield data derived from chamber experiments to estimate the SOC and SOA contributions from several precursors (Kleindienst et al., 2012; Kleindienst et al., 2007). However, the availability of SOA tracer data are limited to only a small number of common precursors, leading to a bias in the quantification of SOC mass (Cheng et al., 2021).

Compared to the above-mentioned approaches, our research group has recently developed a novel Bayesian inference (BI) method that allows for the quantification of POC and SOC based on measurement data of PM major components (i.e., OC, EC, and major secondary inorganic ions). This approach provides better agreement with tracer-based PMF results than traditional techniques. The principle of this approach can be found in Liao et al. (2023). Briefly, our method differentiates POC and SOC by incorporating prior knowledge and measurement data of major PM components to make probabilistic inferences about the unknown POC and SOC mass. This is achieved by considering the parameters as random variables rather than constants, which distinguishes it from the multiple linear regression model. This innovation in methodology offers the potential for quantifying POC and SOC with higher accuracy and lower computational complexity.

Hong Kong is located in southern China and is part of the Guangdong–Hong Kong–Macau Greater Bay Area (GBA) economic and business hub. Since the implementation of the Clean Air Plan in 2013 by the Hong Kong Environment Bureau, air pollution in Hong Kong has significantly improved (Hong Kong Environment Bureau, Bureau, 2013). Numerous studies have been conducted to measure variations in chemical speciation components and potential sources over the past years, but are mainly based on 24-hour offline filter measurements (Hu et al., 2010; Cao et al., 2003; Li et al., 2013). As such, information on POC and SOC in $PM_{2.5}$ are unavailable in time scales down to hourly and for continuous and extended duration covering all seasons. The limited temporal resolution of previous studies has prevented the exploration of diurnal variations and rapid evolutionary processes, which are crucial for understanding SOC formation in the ambient environment. Moreover, long-term continuous measurements are limited in Hong Kong, hindering our comprehensive understanding of episodic events occurring under different seasonal synoptic conditions.

In this study, continuous online monitoring of atmospheric $PM_{2.5}$ and carbonaceous components (i.e., OC and EC) was carried out at a regional suburban site in Hong Kong for nearly one year and a half from 16 July 2020 to 31 December 2021. The objectives of this study are threefold: (1) to identify the optimal method setup and to derive hourly POC and SOC using the novel Bayesian inference approach; (2) to characterize variations of POC and SOC at multi-temporal scales, including diurnal, weekday/weekend, and seasonal variations, and to identify factors influencing SOC

formation; and (3) to investigate SOC variations during city-wide high-PM$_{2.5}$ episodes under different seasonal synoptic conditions. The methodology of this work could serve as a valuable guide for other locations with similar monitoring capabilities. The observation-based POC and SOC data and insights gained regarding pollution processes will provide valuable observation constraints for improving air quality models for our region and other locations.

## 2 Methodology

### 2.1 Aerosol sampling and measurement

The aerosol measurements were conducted at the Hong Kong University of Science and Technology Air Quality Research Supersite (HKUST supersite), which is located on the HKUST campus. Detailed description of this site can be referred to our previous papers (Wang et al., 2022b; Li et al., 2022). Briefly, The HKUST supersite is situated on the hillside of Clear Water Bay in the eastern coastal area of Hong Kong (22.33°N, 114.27°E, Figure S1). It is ~17 km north of the city centers and 2.2 km south of the nearby commercial and urban center of Tseung Kwan O. The sampling site experiences outflow from urban areas in the northwest and southwest directions during 35% of the sampling period. This location represents a typical suburban site. Surrounding the sampling site are evergreen broadleaved woods that are known to emit high levels of biogenic volatile organic compounds (VOCs) (Tsui et al., 2009). The site is characterized as low to moderately polluted, with limited local anthropogenic emissions originating from a nearby construction site for a dormitory and a small canteen. The construction site operates from Monday to Saturday, between 09:00 and 18:00 local time. However, it is important to note that during the study period, the canteen's operations were scaled down to minimum levels due to the ongoing pandemic.

The sampling period lasted for nearly a year and a half from 16 July 2020 to 31 December 2021. Multiple online instruments of hourly or higher time-resolution were deployed to measure the PM$_{2.5}$ levels, its major components as well as gaseous pollutants and meteorological parameters. Briefly, PM$_{2.5}$ mass concentrations were measured by a SHARP monitor (Model 5030i; Thermo Fisher Scientific, USA); major water-soluble inorganic ions (sulfate, nitrate, and ammonium) were monitored by a monitor for aerosols and gases in ambient air (MARGA 1S; Metrohm AG, Switzerland); carbonaceous components (organic carbon, OC, and elemental carbon, EC) were determined by a semicontinuous OC/EC analyzer (model RT-3179; Sunset Laboratory Inc., Oregan, USA). Gaseous pollutants (O$_3$, NO, and NO$_x$) were measured by gas analyzers (Teledyne API 400A, USA; Ecotech Serinus 40, USA, respectively). Meteorological parameters, including temperature, relative humidity (RH), wind speed (WS) and wind direction (WD) were measured by the 10 m automatic weather station (AWS tower, Model 6000, Belfort Instrument Company, USA). The output data from all the above-mentioned instruments were averaged to a resolution of 1-h, and appropriately aligned for the subsequent analysis. Hourly SHARP PM$_{2.5}$ concentrations at the HKUST supersite were corrected due to the measurement bias, see more discussion in Wang et al. (2022b) and Wang et al. (2023) and Text S1 in Supporting Information SI. The PM$_{2.5}$ levels and gas pollutant data in a nearby rural station (Tap Mun, MB) ~15 km to the northeast were used as reference for days before Oct 2020 and after Nov 2021, during which the respective instruments at our sites were either unavailable or under maintenance. The details in treatment for the missing data can be found in our previous studies (Wang et al., 2022b).

**2.2 Estimation of secondary and primary organic carbon by the Bayesian Inference approach**

In this study, the estimation of POC and SOC were performed by the Bayesian Inference approach, which is newly developed in our group (Liao et al., 2023). This method relies on only major chemical composition data that are commonly measured. Specifically, the concentrations of primary and secondary OC are calculated based on Eq. 1-2:

$$OC = EC \times K_1 + SIA \times K_2 \tag{1}$$

$$POC = EC \times K_1, and\ SOC = SIA \times K_2 \tag{2}$$

where OC and EC are the measured hourly concentrations of OC and EC; SIA represents one of the major secondary inorganic ions (i.e., $SO_4^{2-}$, $NO_3^-$, and $NH_4^+$); $K_1$ and $K_2$ are POC/EC ratio and SOC/SIA ratio that are yet to be deduced using Eq. (3).

$$\pi(K_1, K_2|Data) = \frac{L(Data|K_1, K_2)p(K_1, K_2)}{\int L(Data|K_1, K_2)p(K_1, K_2)dK_1 dK_2} \tag{3}$$

where $p(K_1, K_2)$ is the prior distribution of $(K_1, K_2)$, $L(Data|K_1, K_2)$ is the likelihood function of observation data , and $\pi(K_1, K_2|Data)$ is the posterior distribution to be determined. The BI principle is rooted in Bayesian's theorem and embodied in Eq. 3. With the aim of determining the posterior distribution, we first find out the likelihood function $L(Data|K_1, K_2)$ of observation data with respect to parameters to estimate and the prior distribution of such parameter. Eq (4) gives the likelihood function in our Bayesian model by assuming a normal distributed error term.

$$OC \sim N(EC \times K_1 + SIA \times K_2, \sigma_{EC}^2 \times K_1^2 + \sigma_{SIA}^2 \times K_2^2 + \sigma_{OC}^2) \tag{4}$$

Where $\sigma_{EC}$, $\sigma_{OC}$, and $\sigma_{SIA}$ are the uncertainties for EC, OC, and SIA, respectively. The prior distributions of $K_1$ and $K_2$ are set following Eq. 5, signifying the prior knowledge of these two ratios before analyzing the observation data.

$$K_1 \sim N(2.0, 1.0^2), and\ K_2 \sim N(0.4, 0.2^2) \tag{5}$$

Liao et al. (2023) recommend setting these two prior distributions fairly wide to avoid unnecessary constraints, and according to their sensitivity analysis, the influence from different prior distributions becomes negligible when there are adequate observation data. Based on our experience of applying Bayesian inference to estimate POC and SOC, when there are around 10 or more observations in one dataset, estimated posterior distribution of $(K_1, K_2)$ will be robust enough regardless of the prior distribution. Given that the posterior distribution of $(K_1, K_2)$ cannot be solved analytically, we resort to Markov Chain Monte Carlo (MCMC) sampling for numerical estimation, where we construct a Markov chain whose limit distribution is the same as the posterior distribution of interest. The mean values of $(K_1, K_2)$ from such sampling are then used to deduce POC and SOC using Eq. 2.

Finally, from basic error propagation analysis, we further define the uncertainties of POC and SOC (i.e., $\sigma_{POC}$ and $\sigma_{SOC}$) as per Eq. 6:

$$\sigma_{POC} = POC \times \sqrt{(\frac{\sigma_{EC}}{EC})^2 + (\frac{\sigma_{K_1}}{K_1})^2}, and\ \sigma_{SOC} = SOC \times \sqrt{(\frac{\sigma_{SIA}}{SIA})^2 + (\frac{\sigma_{K_2}}{K_2})^2} \tag{6}$$

Compared to other statistical methods (i.e., MIN, MRS, and MLR methods), the BI method allows greater flexibility in model establishment and comprehensive consideration of all measurement uncertainties.

### 2.3 Aerosol liquid water content and acidity estimation

The aerosol water content (AWC), and acidity (pH) were calculated by the thermodynamic equilibrium model ISORROPIA II (http://nenes.eas.gatech.edu/ISORROPIA). The calculation is performed based on the assumption that the aerosol is in metastable state and at chemical equilibrium between the aerosol and gas phase. The model is set in forward mode, with the inputs from MARGA measured species of aerosol phase $Na^+$, $K^+$, $Mg^{2+}$, $Ca^{2+}$, $NH_4^+$, $NO_3^-$, $SO_4^{2-}$, gas phase HCl, $HNO_3$, $NH_3$, ambient temperature, and RH. Detailed information and validation of the model calculation were presented in Text S2 in Supporting Information SI.

## 3 Results and discussion

### 3.1 Determination of POC and SOC by the BI method

Considering the emission sources and secondary formation processes would vary from season to season, we quantified POC and SOC in each individual season to account for seasonal variations. The division of seasons was based on upper-level wind direction, sea-level pressure, and dew point (Yu, 2002; Wong et al., 2022), as shown in Figure S4. Specifically, the seasons were 2020 summer (16 July-28 September 2020), 2020 fall (8 October-23 November 2020), 2020-2021 winter (24 November 2020-28 February 2021), 2021 spring (1 March-2 May 2021), 2021 summer (3 May-7 October 2021), 2021 fall (8 October-24 November 2021) and 2021 winter (25 November-31 December 2021). Within each season, the data was further divided into 24 (hrs) × 4 (wind directions) groups by hour and wind direction (i.e., northwestern, northeastern, southwestern, southeastern) to account for the diurnal and wind direction variations. The BI model was then applied to each group of data. The BI method's inherent advantage allows for this fine division of measurement data, which ensures relatively constant $K_1$ and $K_2$ within each group.

Regarding the selection of the optimal SIA species for tracking SOA in the BI method, Liao et al. (2023) found that the simulation results using sulfate or ammonium as the SOC tracer yielded better agreement with the reference results compared using nitrate. This discrepancy could be attributed to larger measurement artifacts associated with evaporation loss of nitrate in the offline measured data set. For the online measurements, the simulation results using nitrate were also inferior to those using sulfate or ammonium, as the formed SOA is largely retained in the fine particles, while nitrate has propensity to partition onto coarse particles. To further determine the optimal SOC tracers, an uncertainty analysis utilizing an error estimation method was conducted. As shown in Figure S5a, the absolute concentration and uncertainty of POC were highly correlated and showed minimal difference between BI-$SO_4^{2-}$ and BI-$NH_4^+$, whereas the uncertainties for SOC were much larger by BI-$SO_4^{2-}$ than BI-$NH_4^+$. The relative uncertainty, calculated as the uncertainty divided by the concentration, yielded comparable values for POC with both indicators (Figure S5b), while BI-$NH_4^+$ generated larger relative uncertainties for SOC compared to BI-$SO_4^{2-}$. This suggests that sulfate is a better tracer for quantifying SOC levels in our dataset. This conclusion is further supported by a statistical criterion, the Bayesian Information Criterion (BIC), which is universally used in model selection. Lower BIC values

indicate better modelling results. The BIC values for the three SOC tracers were calculated for each individual season, with sulfate consistently yielding the lowest value (Table S1). Thus, the BI-derived POC and SOC using sulfate as the SOC tracer are considered to provide more accurate data with lower uncertainties and are consequently adopted in the subsequent analysis. By incorporating the uncertainties of model estimations to data closure, BIC is designed to help evaluate performance among different models applied to the same dataset. In future applications of the BI model, where specific tracers for aerosol sources are unavailable, the use of PMF may not be feasible. In such cases, we recommend relying on BIC as a reliable method to choose the most suitable SOC tracer.

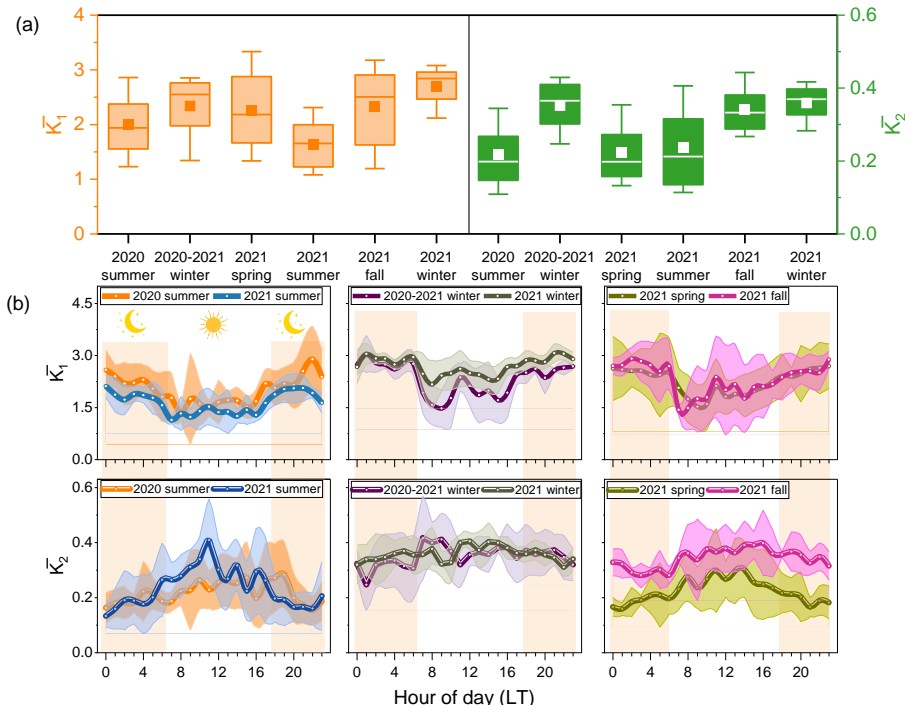

**Figure 1. (a) Box plot of $\overline{K_1}$ and $\overline{K_2}$ values across different seasons (the squares and horizontal lines in the box denote the average and median, the lower and upper boundaries of the boxes represent the 25th and 75th percentile values, and whisker are 10th and 90th percentile). (b) The diurnal variations of $\overline{K_1}$ and $\overline{K_2}$ in individual seasons (solid lines represent the average values, area indicate standard deviation).**

Figure 1 shows the distribution and seasonal variations of $K_1$ and $K_2$ values determined for the $24 \times 4$ groups. The POC/EC ratio ($K_1$) is influenced by the predominant primary sources and meteorological conditions. Due to the monsoon shift in Hong Kong, air pollution during summer is primarily under the control of local emissions, while in cold seasons, regional transport from the continent to the north has a dominant influence. Hence, different predominant sources for POC and EC in Hong Kong are expected. Additionally, many POC species in $PM_{2.5}$ are semi-volatile and are subject to gas-phase partitioning equilibrium, leading to more POC retained in the particle phase at lower temperatures. These factors contribute to a higher $K_1$ values in winter compared to summer (Figure 1a) ($p < 0.01$), and a more pronounced diurnal pattern in summer (Figure 1b). The diurnal variations of K1 in Figure 1b align closely with the local rush hours, during which vehicular emissions exert a dominant influence among all primary sources. In comparison to non-vehicular primary sources, carbonaceous materials originating from vehicular sources exhibit much higher levels of EC, resulting in a lower OC/EC ratio. During periods of heavy traffic, the overall POC/EC ratio

decreases, approaching the typical OC/EC ratio observed for vehicular emissions. On the other hand, SOC/SIA ratio ($K_2$) is influenced by the sources of their precursors, the strength of secondary formation in the atmosphere, and meteorological parameters. Figure 1a shows that $K_2$ is higher (larger mean values) and less variable (smaller inter-quantile ranges) in winter ($p < 0.01$), and Figure 1b demonstrates that hourly $K_2$ values are more stable in winter compared to summer. The long-range regional transport during winter could account for the reduced variability in $K_2$ during cold seasons.

During the periods of 2020-2021 winter and 2021 spring, we have conducted source apportionment analysis using PMF based on a suite of elemental and molecular tracer data in $PM_{2.5}$. The PMF results provide an independent means to determine POC and SOC. Detailed discussions on the wintertime SOC and POC from PMF are documented in our previous publication (Wang et al., 2023). Figure S6 compares POC and SOC estimates obtained from the BI approach versus the PMF method. Good agreements were observed between the two methods for POC in both seasons ($R_p$ = 0.664-0.766). The correlations for SOC simulation showed even stronger agreement with the reference PMF results in winter ($R_p$ = 0.859-0.875). However, in spring, the correlations for SOC exhibited a lower correlation coefficient, and comparable results were obtained when using $SO_4^{2-}$ and $NH_4^+$ as tracers for SOC (Figure S6b). The discrepancy observed in spring could be attributed to the fact that the majority of PMF-resolved SOC was associated with the biogenic secondary organic aerosols factor rather than the secondary sulfate factor.

It is important to highlight that while the BI model demonstrates improved compatibility with PMF results compared to other conventional models, it may not precisely replicate PMF outcomes due to the distinct reaction pathways and formation time spans of SIA and certain SOC components. Considering the similarities in formation pathways, the BI-$SO_4^{2-}$ model would yield more accurate estimations when regional transport has a stronger influence compared to local formation processes. Conversely, when SOC formation pathways are significantly disconnected in time and in space from those of sulfate, the performance of the BI-$SO_4^{2-}$ model would be less satisfactory. For example, in clean regions like the southeast US and Amazon where SOA were dominated by fast local oxidation chemistry of biogenic VOCs (Xu et al., 2015; Riemer et al., 1998; Langford et al., 2022), sulfate may not serve as a good tracer to track SOA in the BI-$SO_4^{2-}$ model. In urban areas where daytime photochemical processing may play a significant role in SOA formation, e.g., summertime Beijing (Duan et al., 2020), sulfate may also fail as a proper tracer. Thus, an integrative evaluation of available PM composition, along with related air pollutant and meteorological conditions, is recommended to aid identification of a suitable SOC tracer in implementing the BI method, as well as assessing the interpretability of the BI method-derived POC and SOC data.

**3.2 Multi-temporal scale variations of POC and SOC**

**3.2.1 Annual levels and seasonal variations**

Figure 2a shows the time series of meteorological parameters, gaseous pollutants, $PM_{2.5}$ and the carbonaceous components, including OC, EC, POC and SOC over the entire measurement period. The study-wide $PM_{2.5}$ concentrations ranged from 1.0 to 94 μg m$^{-3}$ with an average of 14.8 ± 8.8 μg m$^{-3}$. The $PM_{2.5}$ levels varied notably from hour to hour, with 14% exceeding 25 μg m$^{-3}$. This value (25 μg m$^{-3}$) is the new $PM_{2.5}$ annual Interim Target-2

"Air Quality Guidelines (AQG)" recommended by the World Health Organization. It is also the newly proposed Air Quality Objective for $PM_{2.5}$ by the Hong Kong Government (Hong Kong Environment Bureau, Bureau, 2021). The $O_3$ and $NO_x$ concentrations throughout the study period had an average value of $44 \pm 19$ ppb and $8.9 \pm 7.8$ ppb, respectively. The concentrations of OC varied from 0.06-15.7 $\mu gC\ m^{-3}$ (avg. $2.8 \pm 2.0\ \mu gC\ m^{-3}$), and EC ranged from 0.02-6.4 $\mu gC\ m^{-3}$ (avg. $0.76 \pm 0.64\ \mu gC\ m^{-3}$). The average POC was $1.6 \pm 1.3\ \mu gC\ m^{-3}$ (range: 0.06 to 12.4 $\mu gC\ m^{-3}$), approximately two times the average SOC concentration (avg. $0.92 \pm 0.74\ \mu gC\ m^{-3}$, range: 0.02 to 6.8 $\mu gC\ m^{-3}$). Past studies conducted at the same site, using offline measurements of filter samples in 2011-2012 (Huang et al., 2014b) and 2015 (Chow et al., 2022), as well as online measurements during the winter of 2020 (Wang et al., 2023), have also observed higher percentages of POC. The SOC percentage contributions varied under different environments due to the complex sources and formation processes, as well as the meteorological conditions. The contribution of SOC in percentage at our sampling site was lower than those measurements in urban Hong Kong, and other urban cities (Zhou et al., 2014; Zhu et al., 2021; Li et al., 2020), but comparable to a similar suburban site in Shanghai (Wang et al., 2022a; Wang et al., 2022c).

As a sub-tropical region in the southeast coastal of China, the sampling site is under the influence of the seasonal evolution of the East Asian Monsoon system, exhibiting distinctive season-dependent air pollution characteristics. During the summertime, the prevailing wind is from southern oceanic areas, while north-westerlies wind dominated in the winter. Spring and autumn are transitional seasons in between. The RH levels were > 80% in spring and summer, considerably higher than in fall (~70%) and winter (~60%). The wind speeds during winter and fall were higher compared to summer and spring, with the prevailing airflow coming from northwest (Figure 2b). These meteorological conditions would favor the transport and dispersion of air pollutants over larger scale in winter over summer. The seasonal variations in $PM_{2.5}$ showed higher levels in winter and fall compared to spring and summer. $NO_x$ showed the highest levels in the summer and fall of 2021 due to the accumulation of local vehicle emissions from the nearby construction activities, while $O_3$ showed distinct variations with the generally lowest levels in summer. The summer low ozone is attributed to the prevailing southerly flow introduced by the summer monsoon which brings less $O_3$ and /or $O_3$ precursor (So and Wang, 2003). Besides, the strong ozone titration effects by higher $NO_x$ levels could be also responsible for the decreased $O_3$ levels in summer (Zhang et al., 2013). The seasonal variation trends of OC and EC were consistent, with higher concentrations in winter, followed by fall, spring and summer. Similarly, POC and SOC levels were highest in winter and lowest in summer, showing a difference of ~3 times (2.9 vs. 0.8 $\mu gC\ m^{-3}$ and 1.5 vs. 0.5 $\mu gC\ m^{-3}$, respectively) between the two seasons. With regards to the interannual variations, the levels $PM_{2.5}$ and its carbonaceous components exhibited relatively less variations, with comparable levels observed in the same season across different years.

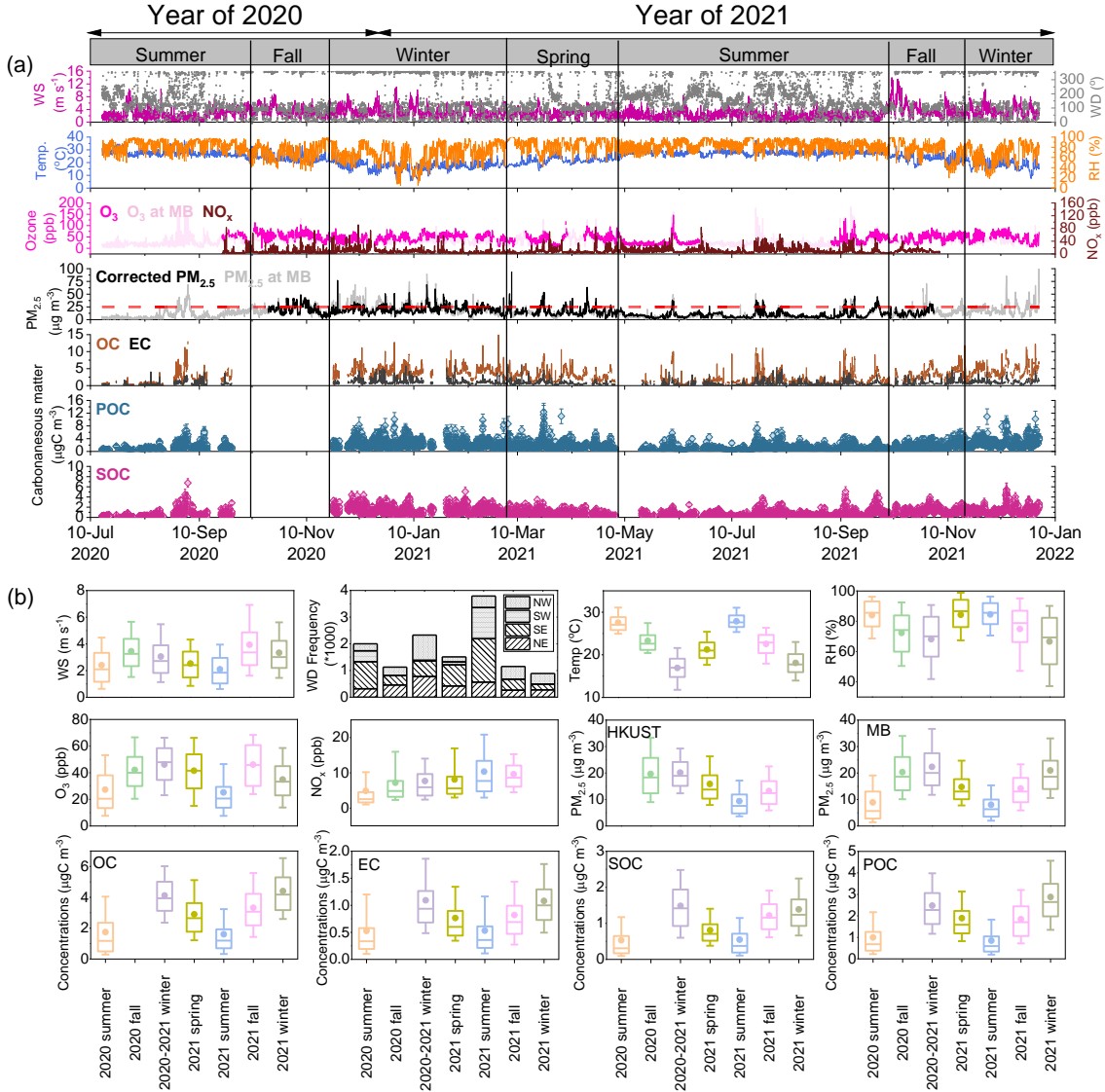

**Figure 2. (a) Time series of meteorological parameters (wind speed, wind direction, temperature, and RH), gaseous pollutants (O₃, and NOₓ), PM₂.₅ (the red dash line marks the WHO AQG IT-4 value), OC and EC, as well as POC and SOC (the *y*-axis error bars represent uncertainties derived from BI method) and (b) Seasonal variations (the circles and horizontal lines in the box denote the average and median, the lower and upper boundaries of the boxes represent the 25th and 75th percentile values, and whisker are 10th and 90th percentile) during the observation period (16 July 2020–31 December 2021) at the HKUST supersite.**

### 3.2.2 Weekend-weekday pattern and diurnal variations

The diurnal variations of PM₂.₅, O₃, NOₓ, and carbonaceous components over the entire period are shown in Figure 3. Since the sampling location is situated near a construction site, we conducted a comparative analysis of diurnal variations between weekdays and weekends to evaluate the influence of construction activities on aerosol particles and gas pollutants. PM₂.₅ displayed minimal disparities between weekdays and weekends, showing flat diurnal cycles across various seasons, except for the winter of 2021. During this particular winter, higher concentrations of PM₂.₅ were observed at night on weekends compared to weekdays. The diurnal variations in O₃ exhibited clear daily trends

throughout different seasons, with higher concentrations during daytime and a peak in the late afternoon. These patterns closely correspond to variations in radiation and temperature. The daily variations of $NO_x$ showed a clear diurnal pattern with higher daytime concentrations on weekdays, which is characterized by two concentration peaks at 9-10 am and 16-18 pm, aligning with the traffic peak hours at the start and at the end of a working day at the construction site. Similarly, EC showed pronounced two peaks during the daytime on weekdays across different

seasons, further indicating the noticeable impacts of primary traffic emissions on $NO_x$ and EC levels, particularly on workdays. Conversely, $NO_x$ and EC levels were generally lower on weekends and lacked a distinct diurnal variation. Different from EC, OC showed less difference between weekdays and Sundays. Slightly higher daytime concentrations with a peak around noon were observed in the two summer seasons, which could be attributed to the enhanced photochemical formation of OC. Diurnal cycles of OC were flatter in other seasons. The higher OC/EC

ratios (Figure S7d) were observed during weekends across different seasons, providing additional evidence of reduced vehicle emissions on non-working days.

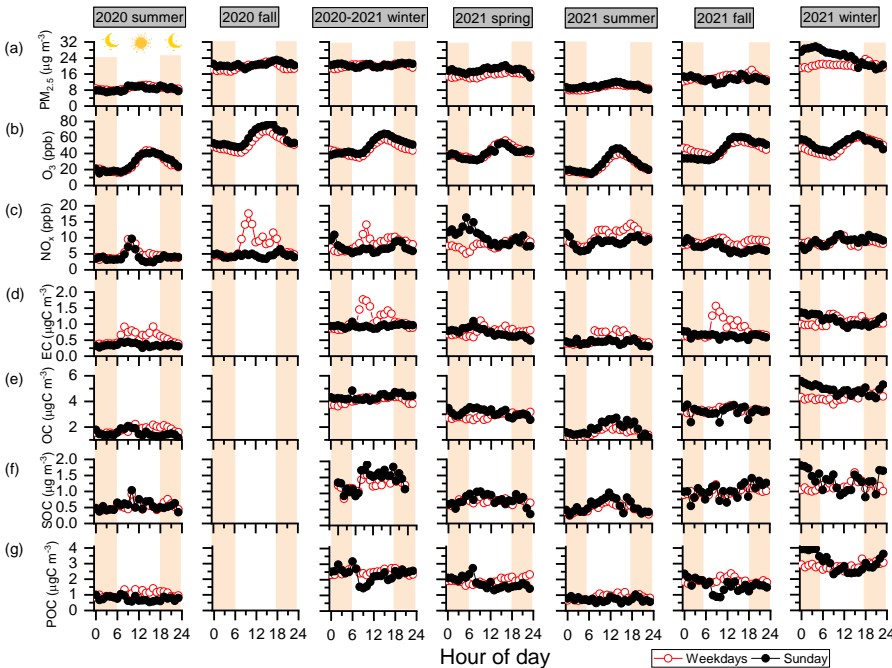

**Figure 3. Diurnal variations of (a) PM₂.₅, (b) O₃, (c) NOₓ, (d) OC, (e) EC, (f) POC and (g) SOC over the entire measurement period. The circles represent the hourly data averaged over weekdays (Monday -Saturday, red) and Sunday (black). The**
**light orange shades represent nighttime periods.**

The weekday-weekend patterns of POC and SOC displayed notable distinctions. Specifically, SOC was slightly higher on weekends, whereas an enhancement of POC was found on weekdays across different seasons. Higher levels of $O_3$ were also observed on weekends, likely due to the weak titration effects as a result of reduced $NO_x$ from vehicle emissions and other anthropogenic emissions during weekends. This observation suggests that anthropogenic

emissions had a stronger influence on POC levels, while SOC levels appeared to be more influenced by the active photochemistry VOCs emissions from the nearby broadleaf woods, rather than the anthropogenic sources. Regarding diurnal variations, POC exhibited comparable levels during nighttime on both weekdays and weekends, while higher

levels were observed during daytime on working days. The much higher POC/SOC ratios during daytime on weekdays (Figure S7e) further collaborated on the impact of primary emissions on POC. Additionally, the SOC levels showed

increased concentrations during daytime on both weekdays and weekends, similar to the daily patterns of $O_3$ as shown in Figure 3b, indicating the influence of photochemical reactions. It should be noted that SOC formation processes are complex and influenced by various factors, including ambient atmospheric oxidants and precursors levels. Moreover, these reactions are significantly influenced by meteorological parameters. Further investigation into the relationships between SOC formation and the aforementioned factors (i.e., temperature, RH, $O_3$ and $NO_x$) will be described in

Sec.3.3.

### 3.3 Characterization of SOC formation dependence on meteorological conditions, $O_x$ and $NO_x$ levels under different pollution conditions

The dataset was divided into five groups based on $PM_{2.5}$ concentrations in individual seasons to investigate the SOC formation under varying pollution conditions. Specifically, periods of $PM_{2.5} < 5$ μg m$^{-3}$ represents the extremely clean

condition; interval of 5 μg m$^{-3}$ < $PM_{2.5}$ < 10 μg m$^{-3}$ represents the clean condition; interval of 10 μg m$^{-3}$ < $PM_{2.5}$ < 15 μg m$^{-3}$, represents the low-pollution condition; interval of 15 μg m$^{-3}$ < $PM_{2.5}$ < 25 μg m$^{-3}$, represents the medium-pollution condition; and interval of 25 μg m$^{-3}$ < $PM_{2.5}$, represents the high-pollution condition. The definition of $PM_{2.5}$ transition value aligns with the annual AQG level, and the Interim Target 2 to 4 limits set by WHO. The evolution of SOC with the increase in the meteorological parameters (i.e., temperature and RH) during different seasons within the

same pollution conditions are shown in Figure 4. SOC concentrations were generally low in low-PM conditions but increased significantly with the intensification of pollution. The highest SOC levels were observed in periods with $PM_{2.5} > 25$ μg m$^{-3}$ during all the seasons, indicating intensive SOC formation contributing to $PM_{2.5}$ air pollution.

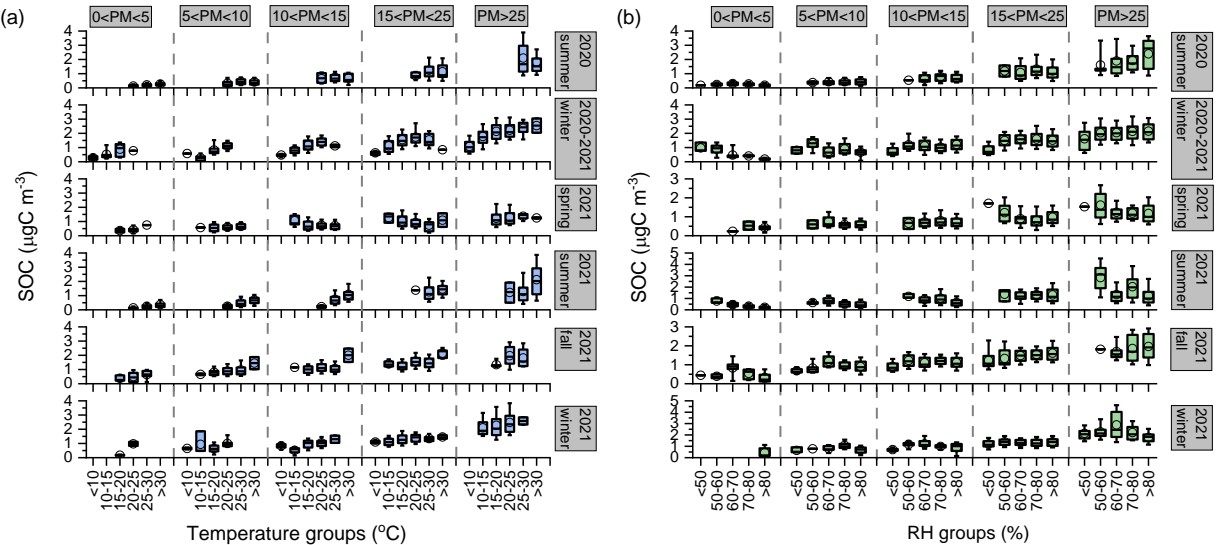

**Figure 4. Concentrations of SOC as a function of (a) temperature bins and (b) RH bins under different $PM_{2.5}$ groups in**
**individual seasons during the entire measurement period (the circles and horizontal lines in the box denote the average and median, the lower and upper boundaries of the boxes represent the 25$^{th}$ and 75$^{th}$ percentile values, and whisker are 10$^{th}$ and 90$^{th}$ percentile).**

As shown in Figure 4a, the average concentrations of SOC were lower than 1 µgC m$^{-3}$ when T < 15°C in all seasons under different pollution conditions, and increased notably with the increasing temperature, reaching the highest levels at T > 30°C. The peak concentrations were almost twice of those in T <10°C. These results highlight the important role of temperature in boosting the ambient SOC formation. Previous chamber experiments and field observations have found that increasing temperature could not only promote the emissions of biogenic VOCs emissions, but also enhance the oxidation reactions (Ding et al., 2011; Svendby et al., 2008). The positive trends were much clear especially in fall and winter during the pollution conditions (PM > 10 µg m$^{-3}$), suggesting that the effects of temperature would be more remarkable on SOC formation in cold seasons. This is further confirmed by the stronger correlation coefficients in winter and fall ($R_p$ = 0.42-0.57) than in spring and summer ($R_p$ = 0.10-0.35).

In contrast to temperature dependence, SOC was less sensitive to RH in all seasons and did not exhibit clear trends (Figure 4b). Under the clean and low PM pollution conditions, SOC showed a much flatter trend with the increasing in RH. At the medium PM pollution conditions, highest SOC levels were observed in low RH groups (RH<60%) and did not change extensively in high RH groups, while under high PM pollution condition, the responses of SOC to RH groups varied across different seasons. In the summer period, SOC levels showed less variations in 2020, while in 2021, highest SOC concentrations occurred in the low RH groups (50%<RH<60%), and then decreased with the increased in RH. The SOC behaviors in the two winter periods were also different, with comparable concentrations across the RH groups in 2020, but much higher SOC levels in the medium RH group in 2021. The SOC behaviors in spring and fall seasons exhibited less variations with the RH groups. The weak relationship between SOC and RH suggested that aqueous phase chemistry may not be the major formation pathway of SOC in our site, which is different from observations in northern China (Guo et al., 2012; Wang et al., 2012) but similar to the location with comparable site characteristics in suburban Shanghai (Wang et al., 2022a).

The relationships between SOC and atmospheric oxidants $O_x$ levels under different pollution conditions in individual seasons are shown in Figure 5a. Atmospheric oxidants $O_x$ ($O_3$+$NO_2$) can be utilized to indicate the ability of atmospheric oxidation associated with photochemical reactions (Kley et al., 1994; Notario et al., 2013). A previous offline measurement study in Hong Kong reported positive correlations between SOC and $O_3$ and highlighted that SOC formation was sensitive to the oxidant levels (Hu et al., 2008). In this study, SOC levels in spring and summer were less sensitive to the $O_x$ under low and medium PM pollution conditions (PM < 25 ug m$^{-3}$). But under the high PM pollution condition, the enhancement of SOC with increase in $O_x$ was only found when $O_x$ < 100 ppb; as $O_x$ further increased, SOC started to decline. In winter, SOC levels showed a clear positive trend with increasing $O_x$, especially under medium and high PM pollution conditions. The daily highest $O_x$ is commonly seen from noontime to late afternoon when the solar radiation is the strongest. The positive correlations of SOC with $O_x$ in winter highlighted that the photochemical formation might remain highly efficient and play an important role in contributing to high SOC levels and air pollution.

The associations of SOC with the $NO_x$ are shown in Figure 5b. Under the clean and medium pollution conditions, the trends between SOC and $NO_x$ were less clear across all the seasons. However, the variations under pollution conditions differed in individual seasons. The concentrations of SOC increased substantially with $NO_x$ in two summer periods

under the pollution conditions, suggesting that $NO_x$ can also be essential to the SOC formation in contributing to photochemical air pollution in summer period (Roberts, 1990; Fan et al., 2022). However, during the winter seasons, when the air masses were dominated by the long-ranged air masses originated from northern China, SOC levels were slightly higher in low $NO_x$ groups than those in high $NO_x$ groups. It's noted that the variations of SOC with $NO_x$ were distinct from those of SOC with $O_3$ in summer and winter seasons, especially under high pollution conditions. These results suggest that the SOC formation pathways are different and might be promoted by various oxidants under different ambient environments.

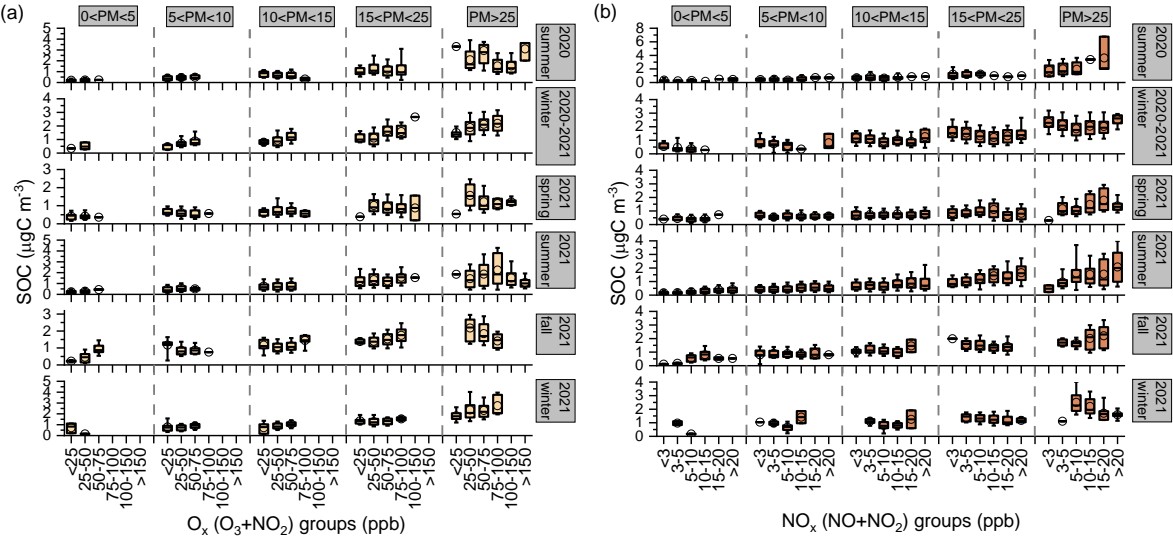

**Figure 5. Concentrations of SOC as a function of (a) $O_x$ bins (b) $NO_x$ bins under different PM2.5 groups in individual seasons during the entire measurement period (the circles and horizontal lines in the box denote the average and median, the lower and upper boundaries of the boxes represent the 25th and 75th percentile values, and whisker are 10th and 90th percentile).**

### 3.4 Evolution characteristics of SOC during the city-wide PM2.5 episodes

#### 3.4.1 Classification of city-wide PM2.5 episodes

As mentioned above, high SOC levels were observed under pollution conditions with PM2.5 higher than 25 µg m$^{-3}$. We further extract the pollution hours to examine the SOC features for gaining insights into its formation mechanisms. We have identified a total of 65 pollution episodes based on a screening method using city-wide air pollutant data from 15 general monitoring stations operated by Hong Kong Environment Protection Department (HKEPD) rather than a single site. The site characteristics and geographical locations of the monitoring stations are shown in Figure S8. The details of this method can be found in our previous publication (Wang et al., 2022b). In this work, PM2.5 episodes were identified as periods of hourly concentrations exceeding 25 µg m$^{-3}$ and lasting 6 consecutive hours or longer at more than three monitoring stations. Figure S9 shows the average concentrations of PM2.5 across the 15 stations and HKUST during individual episodes and those during the non-episode hours. The statistical summary of episode information, PM2.5 averages, the meteorological conditions, and the gas pollutants ($O_3$ and $NO_x$) during individual episodes are listed in Table S2.

As expected, pollution episodes occurred more often in winter (i.e., 24 in 2020-2021 winter, and 7 in 2021 winter, respectively) than in summer and fall. This can be attributed to the less wet deposition and elevated contributions from regional transport, which could be further confirmed by the higher wind speed. We also observed 14 episodes in 2021 spring, possibly due to the dust storms from outside Hong Kong (Ding et al., 2005; Wang et al., 2004). The city-wide $PM_{2.5}$ max-to-min ratios of individual episodes are calculated to investigate the spatial variations (Table S2). A ratio close to 1 indicates the episodic pollution was spatially homogeneous in Hong Kong; a higher value means higher spatial heterogeneity of the episodes across Hong Kong. Generally, the ratios in summer and fall episodes were lower than 2, while higher ratios were observed in winter and spring episodes, indicating the spatial gradient was more notable under the regional influences.

**3.4.2 Variations in SOC during the episodes**

The average concentrations of gaseous pollutants, $PM_{2.5}$, POC and SOC are shown in Figure 6a-e. In general, the concentrations in non-episode hours were higher in winter and fall than those in spring and summer, suggesting the consistent influences of regional transport. Higher concentrations of $O_3$ were observed in winter and fall episodes, with episodes-average of 29-78 ppb, except EP11, EP20 and EP52. It's noted that $NO_x$ levels in these three episodes were much higher than the others. Summer episodes had significantly higher levels, with average concentrations that were more than 2 times than the other seasons. The mass increment ratio (MIR) is calculated as the mass concentration during the episode divided by that during the non-episode hours in the same season, which could be used as an indicator to evaluate the change in the concentration during the episode. The MIR values of $O_3$ and $NO_x$ were generally larger than 1 during the summer and spring episodes (Figure 6g), while close to 1 during winter. In contrast to the gaseous pollutants, the average concentrations of episodic $PM_{2.5}$ showed less seasonal variations, with slightly lower values in 2021 fall. The MIR values were larger than 1 during the majority of episodes, with the highest values in summer (~2-5) and fall (~1.5-2.2) episodes.

The concentrations of carbonaceous components were much higher during episodes than non-episode hours in the same seasons. POC levels were higher in winter episodes, while SOC showed enhancement across different seasons except spring (Figure 6d-e). MIR values >1 were observed for both POC and SOC during summer episodes, with noticeably larger MIR values exceeding 2. Lower MIR values were found in winter seasons, possibly due to the high background levels. The percentage contributions of POC and SOC during individual episodes are shown in Figure 6f. The SOC mass fraction varied in individual seasons, ranging from 10% in EP43 to 66% in EP09. Compared with the corresponding values in non-episode hours, higher SOC percentages were generally found in winter and summer episodes, while they were similar in fall and spring episodes. It is worth noting that the meteorological conditions and major atmospheric oxidants are different in the two seasons, indicating that the formation mechanism differs in the two seasons. Further examination of SOC variations during summer and winter episodes would enhance understanding of SOC formation mechanisms, which will be described in the subsequent sections.

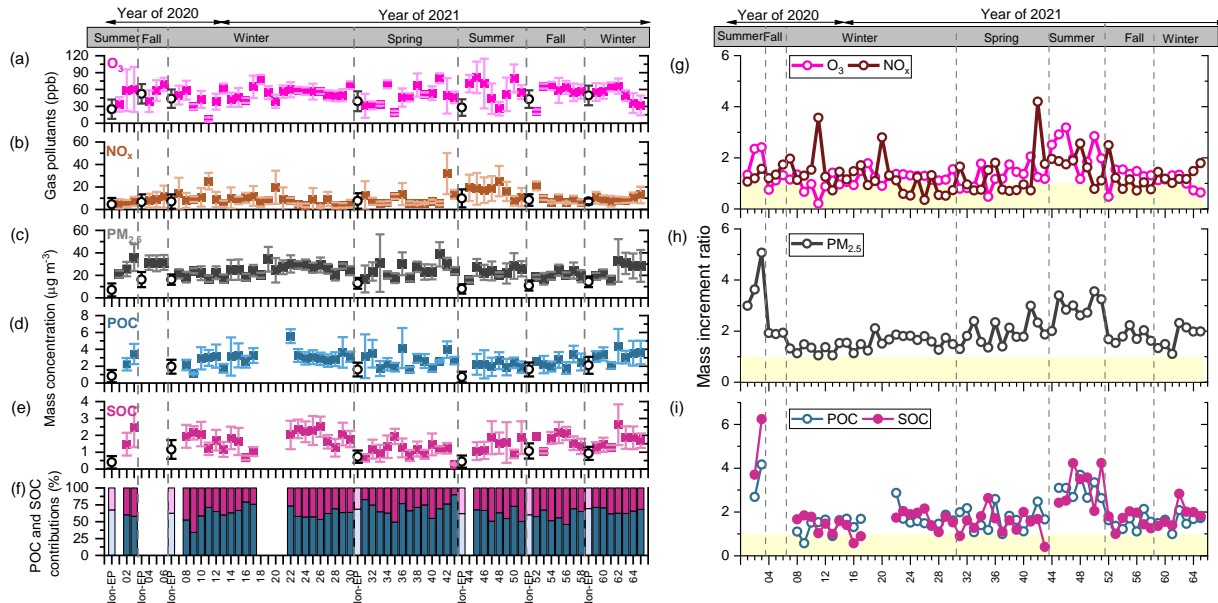

**Figure 6. Comparison of select pollutant levels during episodes and non-episodes for individual episodes. The comparison parameters include concentrations of (a) O₃, (b) NOₓ, (c) PM₂.₅, (d) POC, and (e) SOC, (f) POC and SOC percentage contributions, and mass increment ratios of (g) O₃ and NOₓ, (h) PM₂.₅, and (i) POC and SOC. In panels (a)-(e), the filled squares represent the average values during-episode concentrations while the empty circles represent the average of all non-episode hours throughout the individual season, the error bars represent one standard deviation of the results. In panels (g)-(i), the light-yellow shaded zone marks the mass increment ratios (calculated as mass concentration during the episode divided by that during the non-episode hours in the same season) values of less than 1.**

### 3.4.3 Summer tropical cyclone-induced episodes

A total of 8 episodes were observed during the summer of 2021, with 7 of them coinciding with the presence of tropical cyclones. These cyclones included Typhoon Chaiwan (EP45), Typhoon Infa (EP46 and EP47), Tropical Storm Lupit (EP48), Typhoon Chanthu (EP49 and EP50), and Typhoon Mindule (EP51). The tracks of individual tropical cyclones are shown in Figure S10. It is noted that these tropical cyclones were located east of Hong Kong (near Taiwan). Previous studies have indicated that when a tropical cyclone is situated to the east of Hong Kong, the weather in the region is predominantly influenced by subsidence, resulting in stable air conditions near the ground (Chow et al., 2018; Huang et al., 2006). As shown in Figure 7, the winds during EP45 to EP51 were characterized by low speeds (< 2 m s⁻¹) and come from multiple directions. These stagnant conditions could suppress the vertical dispersion, leading to the accumulation of air pollutants. Episodic PM₂.₅ concentrations show a less spatial gradient with an average max-to-min ratio (~1.6, Table S2). Slightly higher concentrations were observed in new town and urban monitoring stations than suburban sites (Figure S9), in consistent with impacts from local urban sources. This suggests that the air pollution during these episodes was likely attributed to local emissions rather than regional transport.

Concurrent enhancements of gas pollutants and PM₂.₅ mass loadings were observed during the episodes (Figure 7a). PM₂.₅ concentrations notably increased with typhoon evolution, reaching peak values of nearly 50 μg m⁻³ except for EP48. The highest O₃ levels were observed at noon under the influence of Chaiwan and Chanthu2 (EP45 and EP50, respectively). Extremely low NOₓ levels were observed in EP50 with an average of 7.7 ppb. POC and SOC levels

largely increased during the episodes, with different responses in individual episodes. Specifically, a sharp increase in POC was observed in EP45, but SOC levels did not increase noticeably. Similar results were observed in EP46, EP48 and EP50, with the percentage contributions of SOC ranging from 27 to 37%. In contrast, an opposite trend was found in EP47, EP49 and EP51, where SOC exhibited rapid increases, showing higher percentage contributions of 37-50%. It is worth noting that even under the influence of the same typhoon (i.e., EP46 vs. EP 47, EP49 vs, EP50), SOC exhibited distinct variations. These could be associated with the moving tracks of typhoon and its relative location with respect to Hong Kong.

The relationships of SOC with meteorological parameters and oxidants are investigated during the daytime and nighttime episodic hours. Clear diurnal patterns with higher daytime concentrations were observed (Figure 7b), which could be associated with high emissions and strong atmospheric oxidation capacity during the daytime. Positive correlations were found between the logarithm of SOC and 1000/T in both daytime and nighttime episodic hours, with comparable coefficient ($R_p$ = 0.28-0.31, Figure S11a). SOC levels were comparable during the daytime episodic hours among different RH bins, while during nighttime, we observed the increase of SOC with RH raised from 60% to 90%, which might be associated with the aqueous phase reactions.

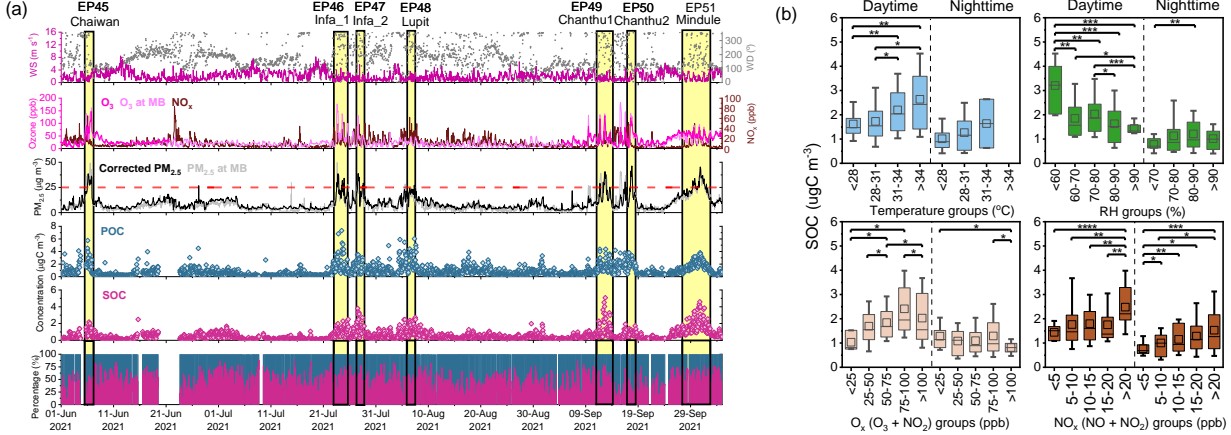

**Figure 7. SOC variation characteristics during typhoon episodes in summer 2021. (a) Time series of meteorological parameters (wind speed and direction), gaseous pollutants (O₃ and NOₓ), PM₂.₅ mass concentrations, POC and SOC levels and their relative percentage contributions, with the yellow shadow area marking individual episode periods of EP45-51. (b) Concentrations of SOC as a function of temperature, RH, Oₓ (O₃+NO₂) and NOₓ bins, with daytime and nighttime episode hours plotted separately (the squares and horizontal lines in the box denote the average and median, the lower and upper boundaries of the boxes represent the 25th and 75th percentile values, and whisker are 10th and 90th percentile. Significance level (p) by t-test: ****p < 0.0001, ***0.0001 < p < 0.001, **0.001 < p < 0.01, *0.01 < p < 0.05).**

The responses of SOC to the oxidants are distinctive in the daytime and nighttime episodic hours. We observed the gradual increase of SOC when Oₓ levels <150 ppb during the daytime, but no clear trend was observed during the nighttime; this could be explained by the negligible photooxidation reactions. Previous study also observed better correlations of SOC with Oₓ in urban Hong Kong during the daytime (Zhou et al., 2014). SOC levels elevated more rapidly with the increase in NOₓ than O₃ in both daytime and nighttime episodic hours. The average levels of SOC were double under conditions of NOₓ levels exceeding 20 ppb compared to the lowest NOₓ group, indicating that NOₓ played a more important role in SOC formation.

### 3.4.4 Winter haze episodes

In winter, $PM_{2.5}$ episodes mainly occurred in December. During the winter episodes, northerly winds prevailed, and the wind speed generally exceeded 3 m s$^{-1}$ (Figure 8a). The city-wide $PM_{2.5}$ showed a clear spatial gradient with an average max-to-min ratio (1.6-4.2, Table S2). Higher levels observed at sites in the northwestern part of the city, followed by the central sites and eastern/southern sites (Figure S9). This spatial pattern is consistent with that wintertime air pollution in Hong Kong is frequently associated with regional transport coming from the north. Notably, the levels of $PM_{2.5}$ were higher during the episodes in 2021 (EP10-13) than those in 2020 (EP62-65). This can be attributed to the increased intensity of anthropogenic emissions in 2021, as the pandemic restrictions in China were somewhat relaxed compared to 2020.

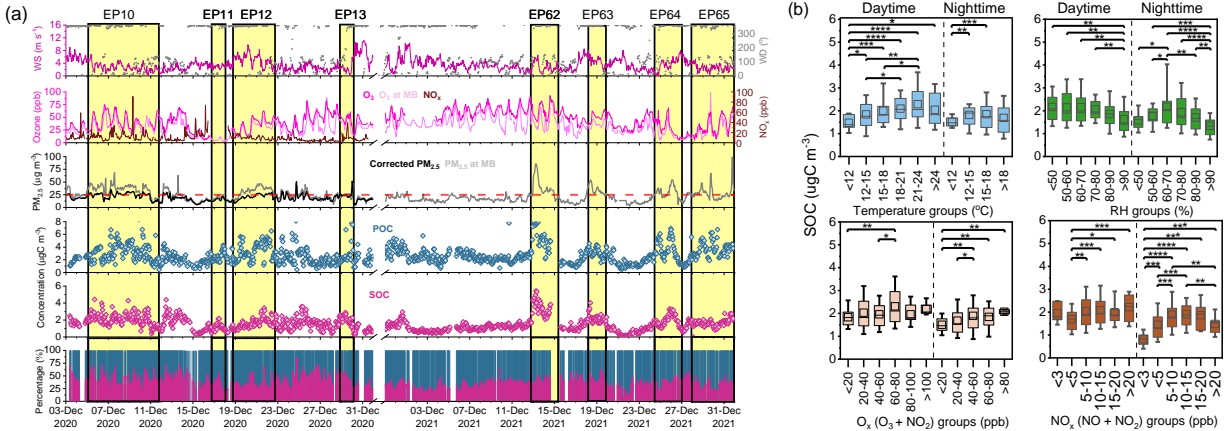

**Figure 8. SOC variation characteristics during haze episodes in winter 2020 and 2021. (a) Time series of meteorological parameters (wind speed and direction), gaseous pollutants ($O_3$ and $NO_x$), $PM_{2.5}$ mass concentrations, POC and SOC levels and their relative percentage contributions, with the yellow shadow area marking individual episodes (EP10-13 and EP62-65). (b) Concentrations of SOC as a function of temperature, RH, $O_x$ ($O_3+NO_2$) and $NO_x$ bins, with daytime and nighttime episode hours plotted separately (the squares and horizontal lines in the box denote the average and median, the lower and upper boundaries of the boxes represent the 25$^{th}$ and 75$^{th}$ percentile values, and whisker are 10$^{th}$ and 90$^{th}$ percentile. Significance level (p) by t-test: \*\*\*\*p < 0.0001, \*\*\*0.0001 < p < 0.001, \*\*0.001 < p < 0.01, \*0.01 < p < 0.05).**

Regarding POC and SOC, we observed enhancements of POC by 1.5-2 times during episodes compared to non-episode hours in the winter of 2020, however, SOC did not exhibit obvious elevation during these episodes. In the winter episodes of 2021, there was concurrent rapid increase in POC and SOC throughout the progression of the episodes. The highest SOC levels were observed in EP62, coinciding with the highest $O_3$ recorded during that episode (Figure 8a). The percentage contributions of SOC varied among individual episodes, ranging from 29% to 41% in 2020 episodes and 32 to 38% in 2021 episodes.

Unlike the summer episodes, the winter episodes exhibited weak diurnal differences, in line with the regional source origin of the pollution, which persisted day and night. Positive correlations of SOC with temperature were only observed during daytime hours while the correlations were insignificant at night. Similar to summer typhoon episodes, the SOC levels during the nighttime in winter initially increased with RH for the three lowest RH bins from < 50% to 70%, then decreased as RH further increased. It is noted that the correlation of SOC with $O_x$ at nighttime was particularly significant during winter episodic hours when the prevailing northly winds were dominant. The average

nocturnal $O_3$ and $O_x$ levels at the sampling site reached around 40 ppb during the nighttime hours. Similar nocturnal $O_3$ enhancements events have been widely observed in recent years in multiple locations in China (He et al., 2022; An et al., 2024), including Hong Kong (Feng et al., 2023). The enhanced levels of nocturnal $O_3$ at the sampling site can increase the ambient oxidation capacity by facilitating the formation of nitrate radical (Brown and Stutz, 2012),

thereby promoting the generation of secondary pollutants. The correlations of SOC with $NO_x$ during the nighttime ($R_p$ = 0.24) was slightly higher than daytime hours ($R_p$ = 0.12), highlighting a notable role of nighttime $NO_x$ chemistry in SOC formation. Previous studies have suggested that nighttime secondary formation is dependent on $NO_3$ radical (Nah et al., 2016; Zhang et al., 2015). Feng et al. (2022) measured nocturnal $NO_3$ radical in Beijing and found that nighttime SOC formation was sensitive to the $NO_3$ radical, providing more direct evidence for the role of $NO_x$ through

enhancing $NO_3$ radical during nighttime SOC formation. If we use $[NO_2][O_3]$ as a rough indicator for the production of $NO_3$ radical (Wang et al., 2018), the observed moderate correlation between nighttime SOC with $[NO_2][O_3]$ ($R_p$ = 0.36) in this study implies that SOA formation pathways involving $NO_3$ radials were also active at our site. Previous studies have also pointed out that SOA formation in cloud and aerosol water played a more important role to total SOA amount, especially in regions of high RH condition (Ervens et al., 2011; Lim et al., 2010). As AWC and acidity

are the major factors for aqueous phase reactions (Jang et al., 2002; Jang et al., 2004), we investigated the relationship between SOC and AWC, as well as aerosol acidity. Table S3 tabulates the average AWC and $[H^+]$ levels during episodic hours, calculated separately for daytime and nighttime, showing higher AWC and acidity during the nighttime episodic hours. Figure S12a shows moderate correlations of SOC with AWC and acidity during nighttime ($R_p$ = 0.30 and 0.35, respectively). The correlations during daytime were less significant ($R_p$ = 0.10 and 0.11, respectively), indicating that

aqueous-phase reactions were negligible during the day. The results, along with our analysis, indicate that both $NO_3$ chemistry and acid-catalyzed aqueous-phase reactions may represent notable formation pathways for nighttime SOC during winter haze episodes.

**4 Conclusions**

Organic carbonaceous aerosols play a significant role in formulating policies to control $PM_{2.5}$ pollution given their

increasing relative contribution to $PM_{2.5}$ in the ambient environment. Availability of POC and SOC from observation-based measurements is crucial for refining atmospheric models and developing more effective measures to tackle carbonaceous aerosol pollution and its associated impacts on climate change and public health. In this study, online observation of major components of $PM_{2.5}$ for *ca*. one and a half years was conducted from 16 July 2020 to 31 December 2021 at a regional suburban site in Hong Kong. POC and SOC were differentiated using a novel Bayesian

inference approach, which yielded results that agree well with those derived from the elemental and organic tracer based-PMF method. The model utilizing sulfate as a SOC tracer exhibited the lowest error and Bayesian Information Criterion (BIC) values, making it a more suitable choice compared to other secondary inorganic ions, such as $NH_4^+$ and $NO_3^-$. We study the characteristics of aerosol carbonaceous components, including seasonal cycles, diurnal and weekday/weekend patterns, and the influencing factors (i.e., meteorological parameters and oxidant levels)

contributing to SOC formation under varied PM pollution conditions. Positive correlation between SOC level and

ambient temperature was observed across different seasons. Substantially high SOC levels were associated with increased $O_x$ concentration, especially in winter, highlighting the important role of photochemical reactions even under weak radiation conditions. $NO_x$ was found to be significant in contributing to extensive SOC formation under pollution conditions in summer.

A total of 65 city-wide $PM_{2.5}$ episodes were identified over the entire study period, and the characteristics of POC and SOC varied substantially among the episodes. An in-depth analysis of summer typhoon episodes and winter haze episodes demonstrated the importance of meteorology and oxidant levels on the variations of SOC and the formation processes. During summer typhoon episodes, the increased carbonaceous components were largely influenced by local emissions resulting from impacts of the typhoons. Higher SOC levels were observed during the daytime, likely due to

enhanced oxidation reactions under high temperatures and stronger solar radiation. In winter haze episodes, the diurnal difference was less obvious as the site was influenced by the continuous regional transport of air pollutants from northern China. Notably, the nighttime aqueous-phase reactions involving the $NO_3$ radical were found to play an important role in SOC formation during the episodic hours.

Overall, our findings demonstrate the diverse facilitating factors contributing to aerosol pollution episodes and

highlight the combined influences of meteorology and atmospheric oxidants on SOC formation. These results will be valuable for modelling studies aiming to improve accuracy in evaluating SOC contributions and variations at both city and regional scales. They will also aid the development of target-oriented strategies for air quality improvement.

*Data availability.* The hourly carbonaceous components and other chemical speciation data presented in this study are available from the data repository maintained by HKUST: https://doi.org/10.14711/dataset/WYJQD0 (Yu and Wang,

590    2023).

*Supplement.* Text S1-S2, Figures S1-S12, and Tables S1-S3.

*Author contributions.* Shan Wang: Formal analysis, Investigation, Data curation, Visualization, Writing – original draft, Writing – review & editing; Kezheng Liao: Methodology, Writing – review & editing; Yuk Ying Cheng, Zijing Zhang, Qiongqiong Wang and Hanzhe Chen: Measurement of other major components, Data validation. Jian Zhen Yu:

Conceptualization, Data curation, Project administration, Supervision, Writing – review & editing.

*Competing interests.* The contact author has declared that none of the authors has any competing interests.

*Disclaimer.* The content of this study does not necessarily reflect the views and policies of the HKSAR Government nor does the mention of trade names or commercial products constitute an endorsement or recommendation of their use.

*Acknowledgements.* We thank funding support from the Hong Kong Research Grants Council (R6011-18, 16305418, and C5004-15E), and the Hong Kong University of Science and Technology (VPRDO19IP01). Special thank goes to Mr. Penggang Zheng, Mr. Xin Feng, and Dr Zhe Wang for providing the gaseous pollutant data at the HKUST supersite.

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
