# Peer review of "Bayesian Inference-Based Estimation of Hourly Primary and Secondary Organic Carbon at Suburban Hong Kong: Multi-temporal Scale Variations and Evolution Characteristics during PM2.5 episodes"

_EGUsphere, 2023_

## Author Comment (AC1)

**Response to Review Comments by Anonymous Referee #1 on "Bayesian Inference-Based Estimation of Hourly Primary and Secondary Organic Carbon at Suburban Hong Kong: Multi-temporal Scale Variations and Evolution Characteristics during PM₂.₅ episodes" by S. Wang et al.**

Wang et al. utilize a recently developed new technique in which Bayesian Inference is used to apportion secondary organic carbon (SOC) and primary organic carbon (POC) from a semi continuous OC/EC analyzer and inorganic aerosol for ~1.5 years worth of observations collected at a suburban-rural site near Hong Kong. The potential value in this approach and observations is that the observations are in-situ (no filter collection and potential biases associated) and provide apportionment of organic carbon (OC) to primary vs secondary, which is important for understanding sources and chemistry and providing recommendations for regulations. The authors argue the Bayesian Inference, which is different than the typical approaches used for OC/EC to apportion SOC and POC, is more robust as there is less influence from assumptions.

With these observations, Wang et al. finds that during the 1.5 year study period, most of the OC was from POC. Further, they observe a time of day and day of week influence on POC and SOC, highlighting the sources and/or chemistry that leads to these two components. The authors then investigated the role meteorology on SOC and POC and observed some influence with temperature and RH, depending on time period and/or amount of total aerosol. Further, the authors find different correlations of SOC and POC to other pollutants, potentially highlighting the chemistry or sources of SOC and POC. Finally, the authors look at different influences on SOC and POC--pollution periods, typhoon, and winter.

The paper is of potential interest to the ACP community. However, as written, it sometimes appears either more of a methods paper or a measurements report paper. This is partially that the authors just look at correlations of data without providing further context in what that correlation (or lack thereof) means for POC and SOC observations. Further, there is some questions about the results that need to be further explored/discussed to understand the robustness of the technique. Some of these comments are discussed in more detail below and should be addressed prior to publication.

Major

We thank the anonymous reviewer for the detailed comments. Below is our point-by-point response to each comment, marked in blue.

1) The major concern is the premise that sulfate was identified to be most suitable for an SOC tracer and the SOC apportionment for the following reasons that need to be explored and/or addressed:

a) The gas-phase production of sulfuric acid is slow. Even at extremely high OH concentration (~1e7 molec. cm^3 OH), the lifetime of SO₂ is ~30 hours, which is slower than the typical production rates observed for fresh SOC production. Further, if there are minimal sources of SO₂ in the near-field (e.g., if there is no coal-fire plants, high sulfur diesel, etc.), the combination of this and relatively long provides concern in using sulfate to partition the OC to SOC. Specifically, could it be underestimating the SOC due to the differences in time scale. Could this potentially explain the under performance of the Bayesian Inference method vs PMF during spring (and how does it look for summer and autumn, if possible?).

b) Another important source of sulfate is through aqueous chemistry (typically the most important source of sulfate in continental and polluted regions). As the time and processing of sulfate is different than fast gas-phase production SOC, how is this influencing the results?

c) Due to a) and b), sulfate is more typically considered a "regional" aerosol pollutant and less of a local pollutant. Thus, how much is the partitioning is just assign highly-aged, regional SOC instead of the fresher, rapidly produced SOC?

d) Looking at the paper that describes the method (Liao et al., ES&T, 2023) and the comparisons of the Bayesian Inference partitioning vs the other methods and the results in the SI, the agreement between the Bayesian Inference and PMF is not great, especially in spring. What is leading to this larger disagreement (e.g., have the authors looked at the curvature and where the data is further away from the 1:1 line)? What is the R^2 for these comparisons and

slope? Could it be that there are time periods where other a priori aerosol are needed to help infer SOC instead of always using sulfate? Further, the results in the SI do not quite match the good agreement (though no statistical analysis is provided to support this) in line 223, page 7.

**Response:** We thank the reviewer for their careful evaluation and attention to details. Here, we would like to first clarify that we do not intend to replace the widely accepted PMF model in apportioning OC to primary and secondary sources; nor are we claiming that Bayesian inference (BI) model is capable of producing results as good as PMF. Inherently, PMF is more accurate in differentiating primary and secondary sources as it considers a suite of tracers that are indicative of major sources. Rather, the purpose of our work is to demonstrate that, when the measurement data of source-specific tracers for PMF are unavailable, the BI method is superior over other previous models, namely minimum OC/EC ratio method, minimum R squared method, and multiple linear regression model. In our EST paper (Liao et al., 2023), we have demonstrated model robustness through sensitivity analyses and used case studies to show that the BI method consistently gives the best compatibility with PMF results among all four models that do not rely on tracer measurements. More specifically, the correlation coefficients of the BI model and the PMF results are between 0.85 – 0.93 for both POC and SOC, suggesting good comparability.

We agree with the points (a)-(c) raised by the reviewer. While sulfate and SOC share the common secondary formation nature, the reaction pathways and formation time spans can vary between them. Consequently, sulfate cannot be considered as a perfect tracer for SOC, especially when freshly generated SOC, originally from local sources, make a notable contribution to the overall SOC levels. This conceptual understanding provides an explanation for the discrepancies observed between the BI method and the PMF results. Below text will be added to the revised manuscript to indicate the limitations of using sulfate as the tracer for SOC in the BI method.

In Methodologies:

" It is important to highlight that while the BI model demonstrates improved compatibility with PMF results compared to other conventional models, it may not precisely replicate PMF outcomes due to the distinct reaction pathways and formation time spans of sulfate and certain SOC components. Consequently, the BI model yields more accurate estimations when regional transport has a stronger influence compared to local formation processes. Nevertheless, our previous research (Liao et al., 2023) demonstrates that the BI model consistently produces results that closely align with PMF, with correlation coefficients exceeding 0.85. "

More information including the slope, Pearson's Correlation Coefficient ($R_p$), Mean Fractional Error (MFE) and Mean Fractional Bias (MFB) are included in Figure S2 (new Figure S6 in the revised manuscript).

The following text and revised Figure S2 will also be added to provide more discussion.

"Good agreements were observed between the two methods for POC in both seasons ($R_p$= 0.664-0.766). The correlations for SOC simulation showed even stronger agreement with the reference PMF results in winter ($R_p$ = 0.859-0.875). However, in spring, the correlations for SOC exhibited a lower correlation coefficient, and comparable results were obtained when using $SO_4^{2-}$ and $NH_4^+$ as tracers for SOC (Figure S6). The discrepancy observed in spring could be attributed to the fact that the majority of PMF-resolved SOC was associated with the biogenic secondary organic aerosols factor rather than the secondary sulfate factor."

[Figure]

**Figure R1 (new Figure S6)**. (a) POC and (b) SOC estimations for the winter (24 November 2020 – 28 February 2021) and spring (1 March -2 May 2021) data. Comparisons are made between the PMF and the BI method, using both sulfate and ammonium considered as tracers for SOC. MFE refers to Mean Fractional Error, and MEB refers to Mean Fractional Bias (MFB).

When selecting between sulfate and ammonium as the SOC tracer in the BI method, we introduce Bayesian Information Criterion (BIC) to help make the selection. By consolidating the uncertainties associated with model predictions and data closure, BIC is designed to assess the performance of different models applied to the same dataset. Note that assessing compatibility with PMF as an evaluation metric fails to consider the model uncertainties inherent in the BI model. From Table S1, one can see that during the spring season, BI-SO$_4$ exhibits a lower BIC value than BI-NH$_4$, indicating superior model performance using sulfate as the SOC tracer for the dataset.

We will add the following description of Bayesian information criterion:

> "By incorporating the uncertainties of model estimations to data closure, BIC is designed to help evaluate performance among different models applied to the same dataset. In future application cases in which the lack

of measurements of source-specific tracers nullifies the use of PMF, one can rely on BIC to choose the most suitable SOC tracer in using the BI method to estimate POC and SOC."

e) Part of this relates to the POC > SOC for most of the observations. It is currently unclear if this is due to being near a construction site or mis-apportioning the rapid, fresher SOC to POC. As the authors describe the location being suburban-to-rural, the influence of POC should be lower as there should be lower sources of POC, except for the construction site located right there. If the apportionment can be shown to be robust (e.g., not missing rapidly produced, fresh SOC) then more discussion on the potential influence of the very localized emissions of POC on the measurements and conclusions about SOC vs POC may need to be discussed (e.g., where were the measurements in relation to the construction and did the influence change with wind direction, etc.).

**Response:** Hong Kong is under the influence of Asian monsoon system. The winter monsoon transported air pollutants from the northern inland areas of China and the PRD region to Hong Kong. The HKUST supersite site, located on the east coast of Hong Kong, serves as a representative receptor site for regional/super-regional pollution originating from the north. Previous source apportionment studies of OC based on offline measurements of filter samples at the same site have consistently observed higher POC than SOC levels. For instance, in 2011-2012, POC accounted for an annual average of 82.9% of total OC (Huang et al., 2014), while in 2015, POC accounted for 85% of total OC (Chow et al., 2022). The majority of POC was apportioned to anthropogenic sources such as industrial emissions, coal combustion and biomass burning, all of which are regional primary combustion sources. A recent study using the elemental and molecular tracer-based sources apportionment on bihourly online measurements in 2020 winter (POC=78%) (Wang et al., 2023) also confirmed the prevalence of higher POC compared to SOC. As shown in Figure R2, in the current study, we observed higher POC levels and POC/SOC ratios when the northern wind prevailed, further confirming the regional characteristics of the carbonaceous aerosols at the HKUST supersite.

[Figure]

**Figure R2.** Bivariate polar plot of mean (a) POC concentrations, (b) SOC concentrations and (d) POC/SOC ratio during the entire campaign.

The following sentence will be added in Section 3.2.1 to describe the higher POC over SOC levels at our site.

"The average POC was $1.6 \pm 1.3$ μgC m$^{-3}$ (range: 0.06 - 12.4 μgC m$^{-3}$), approximately two times higher than the SOC concentration (avg. $0.92 \pm 0.74$ μgC m$^{-3}$, range: 0.02 - 6.8 μgC m$^{-3}$). Past studies conducted at the same site, using offline measurements of filter samples in 2011-2012 (Huang et al., 2014) and 2015 (Chow et al., 2022), as well as online measurements during the winter of 2020 (Wang et al., 2023), have also observed higher percentages of POC."

References:

Chow, W.S., Huang, X.H.H., Leung, K.F., Huang, L., Wu, X., Yu, J.Z. (2022). Molecular and elemental marker-based source apportionment of fine particulate matter at six sites in Hong Kong, China. *Sci Total Environ*, 813, 152652. http://doi.org/10.1016/j.scitotenv.2021.152652

Huang, X.H.H., Bian, Q., Ng, W.M., Louie, P.K.K., Yu, J.Z. (2014). Characterization of PM$_{2.5}$ Major Components and Source Investigation in Suburban Hong Kong: A One Year Monitoring Study. *Aerosol and Air Quality Research*, 14, 237-250. http://doi.org/10.4209/aaqr.2013.01.0020

Liao, K., Wang, Q., Wang, S., Yu, J.Z. (2023). Bayesian Inference Approach to Quantify Primary and Secondary Organic Carbon in Fine Particulate Matter Using Major Species Measurements. *Environmental Science & Technology*, 57, 5169-5179. http://doi.org/10.1021/acs.est.2c09412

Wang, S., Wang, Q., Cheng, Y.Y., Chen, H., Zhang, Z., et al. (2023). Molecular and Elemental Tracers-Based Receptor Modeling of $PM_{2.5}$ in Suburban Hong Kong With Hourly Time-Scale Air Quality Considerations. *Journal of Geophysical Research: Atmospheres*, 128. http://doi.org/10.1029/2023jd039875

2) Page 6, Line 182--It is not clear where the 24x4 groups originate from and the meaning/usage of this. It appears that the 24 comes from the 24 hours in the day, but where is the 4 coming from?

**Response:** The 24 groups come from 24 hours and 4 refer to the 4 main wind directions (i.e., northwestern, northeastern, southwestern, southeastern).

In the revised manuscript, the sentence will be rephased to improve the clarity:

> "Within each season, the data was further divided into 24 (hrs) × 4 (wind directions) groups by hour and wind direction (i.e., northwestern, northeastern, southwestern, southeastern) to account for the diurnal and wind direction variations."

3) It is unclear in Figure 2 what "Corrected" $PM_{2.5}$ means. What was corrected and why was it corrected? This needs to be described in detail and provide evidence in why the correction was needed to be applied. Further, with figure 2, the color selection for (b) was confusing between the box and whisker plots and the wind direction. As there is overlap, it appears that it someone needs to be interpreted that wind direction corresponds to the different colors.

**Response:** (a) During the sampling period, the SHARP Monitor measured hourly $PM_{2.5}$ concentrations were biased, evidenced by the measured SHARP $PM_{2.5}$ concentrations lower than the reconstructed speciation data (Figure R3a). The 24-h offline filter measurement was conducted once every six days during the campaign period. The offline filter measurements showed similar temporal variations with the SHARP $PM_{2.5}$ (Figure R3b). Strong correlations were observed between SHARP $PM_{2.5}$ and filter-based $PM_{2.5}$, with $R_p$ value ranging from 0.86 to 0.99 (Figure R3c), allowing us to correct for the bias of the SHARP $PM_{2.5}$ monitor.

We will revise the wording in Section 2.1 to improve the clarity in the revised manuscript:

> "Hourly SHARP $PM_{2.5}$ concentrations at the HKUST supersite were corrected due to the measurement bias, see more discussion in Wang et al. (2022) and Wang et al. (2023) and Text S1 in Supporting Information S1."

> Text S1 The $PM_{2.5}$ correction

> "The hourly $PM_{2.5}$ concentrations measured by SHARP were biased, evidenced by the measured SHARP $PM_{2.5}$ concentrations consistently falling below the reconstructed speciation data (Figure S2a). The magnitude of bias was determined by comparing the SHARP-measured $PM_{2.5}$ with those 24-h offline filter measurements during the campaign period. The SHARP $PM_{2.5}$ showed similar temporal variations with the offline filter measurements (Figure S2b). Strong correlations were observed between SHARP $PM_{2.5}$ and filter-based $PM_{2.5}$, with $R_p$ values ranging from 0.74 to 0.98. The SHARP $PM_{2.5}$ levels were corrected using the linear relationships shown in Figure S2c."

[Figure]

**Figure R3 (New Figure S2).** (a) Time series of hourly reconstructed $PM_{2.5}$ and SHARP measured $PM_{2.5}$. (b) Time series of daily SHARP measured $PM_{2.5}$ and offline gravimetric mass (Teflon). (c) Linear relationship between online and offline filter-based $PM_{2.5}$ fata during the campaign period.

(b) Suggestion taken. The plot of wind direction has been revised. The revised Figure 2b is copied here for easy reference.

[Figure]

**Figure R4 (revised Figure 2).** (b) Seasonal variations of meteorological parameters (wind speed, wind direction, temperature, and RH), gaseous pollutants ($O_3$, and $NO_x$), $PM_{2.5}$ (the red dash line marks the WHO AQG IT-4 value), OC and EC, as well as POC and SOC during the observation period (16 July 2020–31 December 2021) at the HKUST supersite.

References:

Wang, Q., Wang, S., Cheng, Y.Y., Chen, H., Zhang, Z., et al. (2022). Chemical evolution of secondary organic aerosol tracers during high-PM2.5 episodes at a suburban site in Hong Kong over 4 months of continuous measurement. Atmospheric Chemistry and Physics, 22, 11239-11253. http://doi.org/10.5194/acp-22-11239-2022

Wang, S., Wang, Q., Cheng, Y.Y., Chen, H., Zhang, Z., et al. (2023). Molecular and Elemental Tracers‑Based Receptor Modeling of PM2.5 in Suburban Hong Kong With Hourly Time‑Scale Air Quality Considerations. Journal of Geophysical Research: Atmospheres, 128. http://doi.org/10.1029/2023jd039875

4) Page 10, line 293: It is surprising that SOC would be more influenced by biogenics. How far are the measurements from Hong Kong and how often is urban outflow from Hong Kong influencing the region?

**Response:** The HKUST air quality monitoring supersite is located on the hillside of Clear Water Bay on the east coast of Hong Kong. The site is in a low-density residential neighborhood. It is ~17 km north to the centers of Hong Kong, and 2.2 km south to the nearby commercial and urban centers (Tseung Kwan O), with outflow from urban regions (northwest/southwest direction) accounting for 35% of hours during the sampling period, representing a typical suburban site. The region is characterized by subtropical monsoonal climate, with relatively high temperature and humidity, and sufficient solar radiation throughout the year. The condition is suitable for plant growth, leading to an estimated annual biogenic VOCs emission as high as 9.8 kt (Tsui et al., 2009). The overall conditions in Hong Kong would be favoring the formation of secondary organic aerosols. As shown in Figure R5 (new Figure S1 in the revised manuscript), the sampling site is surrounded by evergreen broadleaved woods with high emissions of biogenic VOCs. In our study, SOC was slightly higher on weekends, which is similar to the $O_3$ patterns. The higher $O_3$ levels on weekends could be attributed to the weak titration effects due to the reduced $NO_x$ from anthropogenic emissions during non-working days. Thus, the slightly higher SOC levels on weekends likely reflected the influence of photooxidation of biogenic VOCs emissions from the nearby broadleaf forests rather than the anthropogenic emissions in the sampling site.

[Figure]

**Figure R5 (new Figure S1).** Location of sampling site and the surrounding environment.

The map of the sampling site will be added as Figure S1, and the site description will be rephased as following:

"Briefly, The HKUST supersite is situated on the hillside of Clear Water Bay in the eastern coastal area of Hong Kong (22.33°N, 114.27°E, Figure S1). It is ~17 km north of the city centers and 2.2 km south of the nearby commercial and urban center of Tseung Kwan O. The sampling site experiences outflow from urban areas in the northwest and southwest directions during 35% of the sampling period. This location represents a typical suburban site. Surrounding the sampling site are evergreen broadleaved woods that are known to emit high levels of biogenic volatile organic compounds (VOCs) (Tsui et al., 2009). The site is characterized as low to moderately polluted, with limited local anthropogenic emissions originating from a nearby construction site for a dormitory and a small canteen. The construction site operates from Monday to Saturday, between 09:00 and 18:00 local time. However, it is important to note that during the study period, the canteen's operations were scaled down to minimum levels due to the ongoing pandemic."

To improve the clarity on SOC concentrations and its weekday-weekend pattern, the sentences will be revised as following:

"The weekday-weekend patterns of POC and SOC displayed notable distinctions. Specifically, SOC was slightly higher on weekends, whereas an enhancement of POC was found on weekdays across different seasons. Higher levels of $O_3$ were also observed on weekends, likely due to the weak titration effects as a result of reduced $NO_x$ from vehicle emissions and other anthropogenic emissions during weekends. This observation

suggests that anthropogenic emissions had a stronger influence on POC levels, while SOC levels appeared to be more influenced by the active photochemistry VOCs emissions from the nearby broadleaf woods, rather than the anthropogenic sources."

References:

Tsui, J.K.-Y., Guenther, A., Yip, W.-K., Chen, F. (2009). A biogenic volatile organic compound emission inventory for Hong Kong. Atmospheric Environment, 43, 6442-6448. http://doi.org/10.1016/j.atmosenv.2008.01.027

5) Section 3.3 vs Section 3.4.3 and 3.4.4 are confusing as they appear to highlight different conclusions. More time needs to be used in trying to bring these sections into a more comprehensive story that provides supporting evidence is needed. Section 3.3 is an example where more in-depth discussions are needed (e.g., why there is less correlation of SOC with oxidants).

**Response:** Thanks for the comment. We would like to clarify that Section 3.3 is an overall characterization of SOC formation dependence on influencing parameters, such as temperature, RH, $O_x$ ($NO_2$+$O_3$) and $NO_x$ (NO+$NO_2$) under different $PM_{2.5}$ pollution levels. This comparison is useful to understand the effects of influencing factors on the SOC concentration under various pollution conditions. The results showed that the effects on SOC formation are much more significant under high pollution hours than clean hours. Therefore, in Section 3.4, we further extract the samples with hourly $PM_{2.5}$ levels > 25 $\mu$g m$^{-3}$ to examine the SOC features for gaining insights into its formation mechanisms. As the meteorological conditions and major atmospheric oxidants are different in summer and winter, we further extract their pollution episodes as two cases to compare the relationship between SOC and influencing factors with and without the sunlight. The results were presented in sub-Section 3.4.3 and 3.4.4, respectively. To make it clearer to the referee and readers, the subtitle will be renamed to "3.3 Characterization of SOC formation dependence on meteorological conditions, $O_x$ and $NO_x$ levels under different pollution conditions."

Regarding the weak correlations of oxidants in Section 3.3, it should be noted that the secondary organic aerosol formation in ambient air is complex and varies in ambient environments as the formation pathways could involve multiple oxidants in addition to the numerous VOC precursors. Many studies have shown that the interaction among precursors, oxidants and products are highly complicated and nonlinear (Zhao et al., 2017; Zhu et al., 2021). Furthermore, the formation is also impacted by meteorological conditions (e.g., temperature, RH, radiation, and pressure), which could significantly affect the reaction kinetics and medium (e.g., gas phase vs. aerosol liquid phase) as well as gas-particle partition of the secondary products (Boyd et al., 2017; Takekawa et al., 2003). Therefore, the effects of oxidant ($O_3$+$NO_2$) on SOC levels likely varies with different environmental conditions.

(b) Further, the discussion of the relationship between $NO_x$ and SOC needs further investigation, as either the $NO_x$ is directly from the construction site, $NO_x$ is an indicator of OH, or $NO_x$ could actually inhibit the production of SOC as $NO_x$ generally leads to more volatile gas-phase products and not less-volatile gas-phase products typically associated with SOC. An example of further discussion/investigation is line 454 - 456 in saying that more $NO_x$ = more SOC and that $NO_x$ play a role in SOC formation. What role and why?

**Response:** We agree with the reviewers that SOA formation will be limited under high $NO_x$ conditions, typically in urban environments with significant vehicle emissions, as $NO_x$ generally leads to more volatile gas-phase products. Previous studies also suggested that SOA formation is retarded under high $NO_x$ levels as a result of change in the relative level of oxidants, i.e. OH, $O_3$ and $NO_3$ and/or the branching ratio for the recombination of organo-peroxyradicals, $RO_2$ (Camredon et al., 2007).

In the campaign period, the $NO_x$ levels were generally low, with an average level of ~5-10 ppb across different seasons (shown in Figure 2b). This is consistent with the characteristics of our sampling site, which is located in suburban Hong Kong with limited influences from vehicle emissions. In our study, positive correlations between SOC and $NO_x$ levels were observed under high pollution conditions, especially during the nighttime episodic hours. We further observed positive correlations of SOC levels with [$NO_2$][$O_3$] (indicator for the production of $NO_3$) during the nighttime episodic hours. The results suggested the nighttime $NO_3$ radical chemistry at our sampling site could play a role in contributing to SOC formation. The importance of nighttime $NO_3$ radical oxidation has been documented in many

chamber experiments and field studies (Fry et al., 2011; Rollins et al., 2012; Wang et al., 2020). Positive correlation was also observed between SOC levels and $NO_x$ during the daytime episodic hours in summer. This could be due to the enhancements of $O_3$ formation with the increased $NO_x$ levels. It has been reported that $O_3$ formation rates would increase with the increase of $NO_x$ concentrations, which would further promote SOA formation (Liu and Shi, 2021). This speculation is supported by the good correlation between SOC levels and $O_x$ levels (Figure 7b). As the formation pathways of surface $O_3$ and SOA are numerous and could vary significantly under different conditions, a more definitive description of the role of $NO_x$ in SOC formation is difficult beyond what we have provided in the manuscript (Lines 453-456).

References:

Boyd, C.M., Nah, T., Xu, L., Berkemeier, T., Ng, N.L. (2017). Secondary Organic Aerosol (SOA) from Nitrate Radical Oxidation of Monoterpenes: Effects of Temperature, Dilution, and Humidity on Aerosol Formation, Mixing, and Evaporation. Environmental Science & Technology, 51, 7831-7841. http://doi.org/10.1021/acs.est.7b01460

Camredon, M., Aumont, B., Lee-Taylor, J., Madronich, S. (2007). The SOA/VOC/NO system: an explicit model of secondary organic aerosol formation. Atmospheric Chemistry and Physics, 7, 5599-5610. http://doi.org/DOI 10.5194/acp-7-5599-2007

Fry, J.L., Kiendler-Scharr, A., Rollins, A.W., Brauers, T., Brown, S.S., et al. (2011). SOA from limonene: role of NO3 in its generation and degradation. Atmospheric Chemistry and Physics, 11, 3879-3894. http://doi.org/10.5194/acp-11-3879-2011

Liu, C., Shi, K. (2021). A review on methodology in $O_3$-$NO_x$-VOC sensitivity study. Environ Pollut, 291, 118249. http://doi.org/10.1016/j.envpol.2021.118249

Rollins, A.W., Browne, E.C., Min, K.E., Pusede, S.E., Wooldridge, P.J., et al. (2012). Evidence for NOx Control over Nighttime SOA Formation. Science, 337, 1210-1212. http://doi.org/10.1126/science.1221520

Takekawa, H., Minoura, H., Yamazaki, S. (2003). Temperature dependence of secondary organic aerosol formation by photo-oxidation of hydrocarbons. Atmospheric Environment, 37, 3413-3424. http://doi.org/10.1016/S1352-2310(03)00359-5

Wang, Y., Hu, M., Wang, Y.-C., Li, X., Fang, X., et al. (2020). Comparative Study of Particulate Organosulfates in Contrasting Atmospheric Environments: Field Evidence for the Significant Influence of Anthropogenic Sulfate and NOx. Environmental Science & Technology Letters, 7, 787-794. http://doi.org/10.1021/acs.estlett.0c00550

Zhao, Y.L., Saleh, R., Saliba, G., Presto, A.A., Gordon, T.D., et al. (2017). Reducing secondary organic aerosol formation from gasoline vehicle exhaust. Proceedings of the National Academy of Sciences of the United States of America, 114, 6984-6989. http://doi.org/10.1073/pnas.1620911114

Zhu, S., Wang, Q., Qiao, L., Zhou, M., Wang, S., et al. (2021). Tracer-based characterization of source variations of PM2.5 and organic carbon in Shanghai influenced by the COVID-19 lockdown. Faraday Discussions, 226, 112-137. http://doi.org/10.1039/d0fd00091d

6) Figure 6--the mass increment ratio needs to be defined in the caption so that the reader does not need to try to find it in the main text.

**Response:** Revised as suggested. We will define the mass increment ratio in the caption in Figure 6.

"Figure 6. Comparison of select pollutant levels during episodes and non-episodes for individual episodes. The comparison parameters include concentrations of (a) $O_3$, (b) $NO_x$, (c) PM2.5, (d) POC, and (e) SOC, (f) POC and SOC percentage contributions, and mass increment ratios of (g) $O_3$ and $NO_x$, (h) PM2.5, and (i) POC and SOC. In panels (a)-(e), the filled squares represent during-episode concentrations while the empty circles represent the combination of all non-episode hours throughout the individual season. In panels (g)-(i), the light-yellow shaded zone marks the mass increment ratios (calculated as mass concentration during the episode divided by that during the non-episode hours in the same season) values of less than 1."

7) Section 3.4.3--it was surprising at first to see typhoon related episodes led to higher PM and more stagnant conditions when it was typhoon season and the typhoons did not pass through the site. This should be discussed or rephrased as the title of the subsection and the low winds and the PM$_{2.5}$ seem contradictory.

**Response:** Tropical cyclones are low pressure systems that develop into an intense storm over tropical water. Typhoon is a mature tropical cyclone that develops in northwestern Pacific, with size of 1000-200 km. Typhoons occur frequently in late summer and early autumn every year. When a typhoon is located east of Hong Kong and its center several hundred km away (e.g., near Taiwan), subsidence dominates over Hong Kong, the air near ground become stable (subsidence inversion) (Chow et al., 2018; Huang et al., 2006). Therefore, the weather during approaching typhoons is calm and cloudless with low wind speed, which suppresses the vertical dispersion of air pollutants, leading to higher PM$_{2.5}$ levels and enhanced formation of secondary pollutants. Whereas, when a tropical cyclone is located to the south or west of Hong Kong, winds in Hong Kong would be southeasterly, the air quality in Hong Kong is generally good or normal due to tropical cyclone-associated precipitation, oceanic air mass transport and an enhanced rising motion.

During June to October 2021, over ten tropical cyclones occurred over the western North Pacific, and five of them were located east of Hong Kong, namely Typhoon Chaiwan, Typhoon Infa, Tropical Storm Lupit, Typhoon Chanthu and Typhoon Mindule. Their provisional tracks of these cyclones, along with their corresponding atmospheric pressure readings, are shown in Figure S7 (Figure S10 in the revised manuscript).

To improve the clarity, the title of subsection 3.4.3 will be revised as "Summer tropical cyclone-induced episodes". Besides, the following sentence will be included in the revised manuscript to explain the reasons.

> "A total of 8 episodes were observed during the summer of 2021, with 7 of them coinciding with the presence of tropical cyclones. These cyclones included Typhoon Chaiwan (EP45), Typhoon Infa (EP46 and EP47), Tropical Storm Lupit (EP48), Typhoon Chanthu (EP49 and EP50), and Typhoon Mindule (EP51). The tracks of individual tropical cyclones are shown in Figure S10. It is noted that these tropical cyclones were located east of Hong Kong (near Taiwan). Previous studies have indicated that when a tropical cyclone is situated to the east of Hong Kong, the weather in the region is predominantly influenced by subsidence, resulting in stable air conditions near the ground (Chow et al., 2018; Huang et al., 2006). As shown in Figure 7, the winds during EP45 to EP51 were characterized by low speeds (< 2 m s$^{-1}$) and come from multiple directions. These stagnant conditions could suppress the vertical dispersion, leading to the accumulation of air pollutants. Episodic PM$_{2.5}$ concentrations did not show a clear spatial gradient (Figure S9) with comparable concentrations observed over Hong Kong. This suggests that the air pollution during these episodes was likely attributed to local emissions rather than regional transport."

References:

Chow, E.C.H., Li, R.C.Y., Zhou, W. (2018). Influence of Tropical Cyclones on Hong Kong Air Quality. Advances in Atmospheric Sciences, 35, 1177-1188. http://doi.org/10.1007/s00376-018-7225-4

Huang, J.P., Fung, J.C.H., Lau, A.K.H. (2006). Integrated processes analysis and systematic meteorological classification of ozone episodes in Hong Kong. Journal of Geophysical Research: Atmospheres, 111. http://doi.org/10.1029/2005jd007012

8) Overall statistics. (a) There are many instances where it was stated there was a good correlation between different variables when the Pearson's coefficient was 0.5 or lower, which means that only ~25% of the variation could be capture by the correlation. Thus, much of the discussion about good correlations needs to be softened or rephrased.

**Response:** Thanks for the comment. We acknowledge that the statement of "a good correlation" could be subjective. Here we rephrase the cases when the Pearson's coefficient below 0.6 to "moderately correlated" instead.

(b) Further, it is discussed that there are differences between observations, but looking at the box and whisker plots, the data looks nearly identical. Statistical analysis of the different binned data and including an asterisk for data that is significantly different from other bins would be beneficial.

**Response:** Thanks for the comments. Please see the revised Figure 7b and Figure 8b with the inclusion of asterisk for data that is significantly different from other bins.

[Figure]

**Figure R6.** Left: revised Figure 7b. Right: revised Figure 8b.

(c) As stated above, Fig. S2 needs more discussion and consideration for statistics.

**Response:** More discussion on Figure S2 has been included in the revised manuscript. See our response to comment 1(e).

(d) Finally, it is not clear why the units were selected as they were for Fig. S8 (why ln(SOC), ln(RH), and 1000/T))?

**Response:** Figure S8 are generated based on past practices seen in the laboratory experiments and filed measurements (Shen et al., 2015; von Hessberg et al., 2009; Yuan et al., 2018). The purpose of using 1000/T is that in most temperature-dependent chamber studies, the use of "exp(1000/T)" could potentially resolve the discrepancy and give the most appropriate simulation results (Jenkin et al., 2018). Therefore, we investigate the relationship between ln (SOC) and 1000/T. To keep the unit of x-and y-axis in consistent log unit, we used the ln(RH) instead. Note that Figure R7 compares the correlations of meteorological parameters and oxidants using different units, showing the correlations being quite similar and comparable. To make it simpler and clearer to readers, we will keep using the logarithm of SOC and the 1000/T, whereas using the linear unit for other parameters.

[Figure]

**Figure R7:** Scatter plot of SOC with meteorological parameters (i.e., temperature and RH), and oxidants (i.e., $O_x$ and $NO_x$) during the episodic hours in (a) summer typhoon episodes and (b) winter haze episodes. The solid red circle represents daytime hours, blank green circle represents nighttime hours. Left: logarithm comparison; Right: linear comparison.

References:

Jenkin, M.E., Valorso, R., Aumont, B., Rickard, A.R., Wallington, T.J. (2018). Estimation of rate coefficients and branching ratios for gas-phase reactions of OH with aromatic organic compounds for use in automated mechanism construction. Atmospheric Chemistry and Physics, 18, 9329-9349. http://doi.org/10.5194/acp-18-9329-2018

Shen, R.Q., Ding, X., He, Q.F., Cong, Z.Y., Yu, Q.Q., Wang, X.M. (2015). Seasonal variation of secondary organic aerosol tracers in Central Tibetan Plateau. Atmospheric Chemistry and Physics, 15, 8781-8793. http://doi.org/10.5194/acp-15-8781-2015

von Hessberg, C., von Hessberg, P., Pöschl, U., Bilde, M., Nielsen, O.J., Moortgat, G.K. (2009). Temperature and humidity dependence of secondary organic aerosol yield from the ozonolysis of β-pinene. Atmospheric Chemistry and Physics, 9, 3583-3599. http://doi.org/10.5194/acp-9-3583-2009

Yuan, Q., Lai, S., Song, J., Ding, X., Zheng, L., et al. (2018). Seasonal cycles of secondary organic aerosol tracers in rural Guangzhou, Southern China: The importance of atmospheric oxidants. Environmental Pollution, 240, 884-893. http://doi.org/10.1016/j.envpol.2018.05.009

9) (a) Section 3.4.4--It is unclear if the necessary compounds were measured to discuss acid-catalyzed reactions (e.g., were gas-phase $HNO_3$ and/or $NH_3$ measured, were the cations listed for ISORROPIA measured)? If none of these were measured, acidity of the aerosol should not be considered as the thermodynamic model is under-constrained. If they were measured, comparisons of the measurements, including the fractional contribution of either gas-phase to its total (e.g., gas-phase $HNO_3$ / (aerosol-phase $NO_3$ + gas-phase $NO_3$)) from the model and observations needs to be provided to provide confidence in the acidity calculations.

**Response:** The gas-phase $HNO_3$ and $NH_3$, and particle-phase anions and cations listed as input for ISORROPIA were measured by a monitor for aerosols and gases in ambient air (MARGA 1S; Metrohm AG, Switzerland).

The following underlined text are newly added and will be included in the revised manuscript:

"2.3 Aerosol liquid water content and acidity estimation"

The aerosol water content (AWC), and acidity (pH) were calculated by the thermodynamic equilibrium model ISORROPIA II (http://nenes.eas.gatech.edu/ISORROPIA). The calculation is performed based on the assumption that the aerosol is in metastable state and at chemical equilibrium between the aerosol and gas phase. The model is set in forward mode, with the inputs from MARGA-measured species of aerosol phase $Na^+$, $K^+$, $Mg^{2+}$, $Ca^{2+}$, $NH_4^+$, $NO_3^-$, $SO_4^{2-}$, gas phase HCl, $HNO_3$, $NH_3$, ambient temperature, and RH. Detailed information and validation of the model calculation were presented in Text S2 in Supporting Information S1."

Text S2 Aerosol water content and pH verification

Figure S3 shows the comparisons of predicted and measured $HNO_3$, $NO_3^-$, and $\varepsilon(HNO_3)$, and $NH_3$, $NH_4^+$ and $\varepsilon(NH_4^+)$ for summer and winter episode cases. The gas-particle phase partitioning of $HNO_3$, namely $\varepsilon(HNO_3)$, is described as gas-phase $HNO_3$ concentration divided by the sum of aerosol-phase $NO_3$ and gas-phase $HNO_3$. Similarly, the $\varepsilon(NH_4^+)$ is calculated as aerosol-phase $NH_4^+$ divided by the sum of aerosol-phase $NH_4^+$ and gas-phase $NH_3$.

The partitioning ratios for summer and winter episode cases were as follows: $\varepsilon(NH_4^+) = 49 \pm 18\%$ and $61 \pm 17\%$ for summer and winter, $\varepsilon(HNO_3) = 28 \pm 9.2\%$ and $17 \pm 8.0\%$ for summer and winter, respectively. Good correlations were observed between the predicted and measured gas-phase $HNO_3$, aerosol-phase $NO_3^-$ and $\varepsilon(HNO_3)$, with $R_p$ in the range of 0.56-0.86. Despite the good correlations, ISOROPPIA-II predicted gas-phase $HNO_3$ was systematically higher than measured value, with regression slopes of larger than 2.0. This could be attributed to the underestimate of $HNO_3$ measured by MARGA system (Makkonen et al., 2014). Regarding the aerosol-phase $NO_3^-$, the ISOROPPIA-II underpredicted the values with regression slope of 0.62 and 0.82 in summer and winter, respectively. This leads to a regression slope of higher than 1 comparing predicted to measured $\varepsilon(HNO_3)$. In contrast, predicted versus measured gas-phase $NH_3$, aerosol-phase $NH_4^+$ and $\varepsilon(NH_4^+)$ is close to 1:1 and highly correlated ($R_p$: 0.77-0.96). The overall good correlations between the predicted and measured values of $NH_4^+$, $NO_3^-$, gas-phase $NH_3$ and $\varepsilon(NH_4^+)$ suggest the precision and consistent utility of thermodynamic equilibrium model in the quantification of $PM_{2.5}$ aerosol water content and acidity.

[Figure]

**Figure R8 (new Figure S3).** Comparisons of (a, b, c) predicted and measured $HNO_3$, $NO_3^-$, and $\varepsilon(NO_3^-)$ and (d, e, f) $NH_3$, $NH_4^+$, and $\varepsilon(NH_4^+)$ for summer and winter episode cases.

(b) Further, it is surprising to discuss that there is aqueous-phase nitrate radical chemistry, as the radical has very low Henry's Law constant meaning it does not readily go into the aerosol liquid water. Finally, the last sentence that states that acid-catalyzed reactions with $NO_3$ radical is very surprising as this seems thermodynamically unfavorable. Are there laboratory studies that have observed this? If so, please provide appropriate references.

**Response:** We agree with the reviewer's comment. The revised text is copied here for reference, and will be incorporated in the revised manuscript:

"Previous studies have also pointed out that secondary organic aerosol formation in cloud and aerosol water played a more important role to total secondary organic aerosols, especially in regions of high relative humidity condition (Ervens et al., 2011; Lim et al., 2010). As AWC and acidity are the major factors for aqueous phase reactions (Jang et al., 2002; Jang et al., 2004), we investigated the relationship between SOC and AWC, as well as aerosol acidity. Table S3 tabulates the average AWC and [$H^+$] levels during episodic hours, calculated separately for daytime and nighttime, showing higher AWC and acidity during the nighttime episodic hours. Figure S12a shows moderate correlations of SOC with AWC and acidity during nighttime ($R_p$ = 0.30 and 0.35, respectively). The correlations during daytime were less significant ($R_p$ = 0.10 and 0.11, respectively), indicating that aqueous-phase reactions were negligible during the day. The results, along with our analysis, indicate that both $NO_3$ chemistry and acid-catalyzed aqueous-phase reactions may represent notable formation pathways for nighttime SOC formation during winter haze episodes."

In Abstract

"Enhanced SOC formation was observed with the increase in nocturnal $NO_3$ radical (represented by [$NO_2$][$O_3$]) and under conditions with high water content and strong acidity. This suggests that both $NO_3$ chemistry and

acid-catalyzed aqueous-phase reactions are likely notable contributors to SOC formation during winter haze episodes."

(c) Another concern is that there is correlation of SOC with $O_x$ at nighttime, as $O_3$ should be low at nighttime due to no production. Why is there correlation?

**Response:** The nocturnal $O_3$ enhancement events are widely observed in recent years in mainland China. The reported nocturnal ozone enhancements ranged from 5 to 30 ppb, with mean ozone peak could exceed 40 ppb (An et al., 2024; He et al., 2022). Feng et al. (2023) investigated the diurnal variations of surface ozone from 2011-2022 at 11 monitoring stations in Hong Kong. In addition to daytime $O_3$ peak, elevated nocturnal $O_3$ levels were found, with peak concentration over 20 ppb, which are thought to be driven by enhanced regional transport of $O_3$-rich plumes and weakened NO titration effect.

In our study, we also observed enhanced nocturnal $O_3$, and $O_x$ levels at both the sampling site and regional background site (Tap Mun, MB station) during the summer and winter episodes (Figure R9). The average $O_3$ levels reached around 40 ppb at HKUST during the nighttime episodic hours. The wind rose analyses suggest that during the nighttime (18:00 pm to 06:00 am), the northeasterly and northly wind prevailing in summer and winter episodes, respectively (Figure R10). The horizontal transport of ozone-rich air from the north/northwestern Great Bay Area leads to enhancement in nocturnal $O_3$ levels in Hong Kong. It is noted that the correlation of SOC with $O_x$ at nighttime was much significantly in winter episodic hours when the prevailing northly winds predominant. Previous studies have suggested that the enhanced nocturnal ozone can increase the oxidation capacity, by simulating nitrate radical formation (Brown and Stutz, 2012), and promote the formation of secondary pollutants, such as particulate nitrate and secondary organic aerosols. Thus, the correlations between secondary pollutants and nighttime $O_3$ are expected in ambient environment.

The following sentence will be added in Section 3.4.4 to describe the nocturnal $O_3$ levels and the correlations of $O_x$ with SOC.

"It is noted that the correlation of SOC with $O_x$ at nighttime was particularly significant during winter episodic hours when the prevailing northly winds were dominant. The average nocturnal $O_3$ and $O_x$ levels at the sampling site reached around 40 ppb during the nighttime hours when the prevailing northly wind was dominant. Similar nocturnal $O_3$ enhancements events have been widely observed in recent years in mainland China (An et al., 2024; He et al., 2022) and Hong Kong (Feng et al., 2023). The enhanced levels of nocturnal $O_3$ at the sampling site can increase the ambient oxidation capacity by facilitating the formation of nitrate radical (Brown and Stutz, 2012), thereby promoting the generation of secondary pollutants."

[Figure]

**Figure R9.** Diurnal variations of $O_3$, $O_x$, $NO_x$ at Tap Mun (MB station) and HKUST site, and wind rose during (a) summer episodes and (b) winter episodes.

[Figure]

**Figure R10.** Wind rose during (a) summer daytime episodic hours, (b) summer nighttime episodic hours, (c) winter daytime episodic hours and (d) winter nighttime episodic hours.

References:

An, C., Li, H., Ji, Y., Chu, W., Yan, X., Chai, F. (2024). A review on nocturnal surface ozone enhancement: Characterization, formation causes, and atmospheric chemical effects. Sci Total Environ, 921, 170731. http://doi.org/10.1016/j.scitotenv.2024.170731

Brown, S.S., Stutz, J. (2012). Nighttime radical observations and chemistry. Chem Soc Rev, 41, 6405-6447. http://doi.org/10.1039/c2cs35181a

Ervens, B., Turpin, B.J., Weber, R.J. (2011). Secondary organic aerosol formation in cloud droplets and aqueous particles (aqSOA): a review of laboratory, field and model studies. Atmospheric Chemistry and Physics, 11, 11069-11102. http://doi.org/10.5194/acp-11-11069-2011

Feng, X., Guo, J., Wang, Z., Gu, D., Ho, K.-F., et al. (2023). Investigation of the multi-year trend of surface ozone and ozone-precursor relationship in Hong Kong. Atmospheric Environment, 315. http://doi.org/10.1016/j.atmosenv.2023.120139

He, C., Lu, X., Wang, H., Wang, H., Li, Y., et al. (2022). The unexpected high frequency of nocturnal surface ozone enhancement events over China: characteristics and mechanisms. Atmospheric Chemistry and Physics, 22, 15243-15261. http://doi.org/10.5194/acp-22-15243-2022

Jang, M., Czoschke, N.M., Lee, S., Kamens, R.M. (2002). Heterogeneous atmospheric aerosol production by acid-catalyzed particle-phase reactions. Science, 298, 814-817. http://doi.org/10.1126/science.1075798

Jang, M., Czoschke, N.M., Northcross, A.L. (2004). Atmospheric organic aerosol production by heterogeneous acid-catalyzed reactions. Chemphyschem, 5, 1647-1661. http://doi.org/10.1002/cphc.200301077

Lim, Y.B., Tan, Y., Perri, M.J., Seitzinger, S.P., Turpin, B.J. (2010). Aqueous chemistry and its role in secondary organic aerosol (SOA) formation. Atmospheric Chemistry and Physics, 10, 10521-10539. http://doi.org/10.5194/acp-10-10521-2010

Makkonen, U., Virkkula, A., Hellén, H., Hemmilä, M., Sund, J., et al. (2014). Semi-continuous gas and inorganic aerosol measurements at a boreal forest site: seasonal and diurnal cycles of $NH_3$, HONO and $HNO_3$. Boreal Environment Research, 19 (suppl. B), 311–328.

---

## Author Comment (AC2)

**Response to Review Comments by Anonymous Referee #2 on "Bayesian Inference-Based Estimation of Hourly Primary and Secondary Organic Carbon at Suburban Hong Kong: Multi-temporal Scale Variations and Evolution Characteristics during PM$_{2.5}$ episodes" by S. Wang et al.**

Review of "Bayesian Inference-Based Estimation of Hourly Primary and Secondary Organic Carbon at Suburban Hong Kong: Multi-temporal Scale Variations and Evolution Characteristics during PM$_{2.5}$ episodes" by Wang et al.

General comments

The manuscript focuses on understanding primary and secondary organic carbon (POC and SOC) in PM$_{2.5}$ and their driving factors. The study was conducted in suburban Hong Kong from July 2020 to December 2021. It employs a novel Bayesian inference approach to differentiate between POC and SOC, using sulfate as a tracer for SOC. The study explores the temporal characteristics of POC and SOC, including diurnal, weekly, and seasonal variations, and their evolution during PM2.5 episodes. The methodology developed offers practical guidance for similar studies elsewhere, providing valuable insights for atmospheric models and understanding PM pollution processes. The study's results indicate distinct SOC formations under different seasonal and pollution conditions, influenced by factors like temperature, relative humidity, and atmospheric oxidants. This research contributes to refining atmospheric models and developing strategies for air quality improvement and climate change mitigation. The results are presented effectively, with appropriate statistical analysis and visual aids. This paper is within the scope of ACP and might be of great interest to the broad atmospheric science community.

We thank the anonymous reviewer for the detailed comments. Below is our point-by-point response to each comment, marked in blue. The related text in the manuscript is copied here for reference.

However, there are areas for improvement in terms of clarity and depth in the discussion of certain results, particularly the implications of the findings for broader atmospheric science and policy-making. Currently, it tends to read more as a data-centric measurement report. Further elaboration on the Bayesian inference method used is recommended for accessibility to readers less familiar with this approach. I agree with the concerns raised by Anonymous Referee #1, particularly regarding the use of sulfate as the best tracer for SOC for all seasons and pollution episodes.

**Response:**

Please see our response to comment 1 raised by reviewer #1.

Additionally, I have a few specific questions and comments that should also be addressed before the manuscript can be considered for publication.

Specific comments:

1. Figure 1b: Could you explain the noticeable decrease in K1 observed between 6 and 7 am during winter, spring, and fall?

**Response:** We thank the reviewer for pointing out this missing piece of information. In short, such a drop in K1 can be attributed to the increase of traffic emissions during rush hours. The definition of K1 is POC/EC, which, in other words, is the average of OC/EC ratios from all primary sources weighted by their contributions. It is foreseeable that for vehicular emissions, the OC/EC ratio should be much lower than those from non-vehicular emissions, considering the significantly elevated EC level from engines. According to our PMF results, the OC/EC from vehicular emission is about 0.5, whereas for other primary sources (e.g., industrial emission, cooking emission, residual oil combustion), the OC/EC ratio is 0.85-1.18. Therefore, as the contribution from vehicular emissions to the primary pollution increases, the POC/EC ratio, i.e., K1, is bound to decrease towards the OC/EC of vehicular emission. We add some explanations in this regard.

The following sentence will be added to describe the noticeable decrease in K1 observed between 6 and 7 am.

> "The diurnal variations of K1 in Figure 1b align closely with the local rush hours, when vehicular emissions exert a dominant influence among all primary sources. In comparison to non-vehicular primary sources, the

EC amount from vehicular sources is much higher, resulting in a lower OC/EC. During periods of heavy traffic, the overall POC/EC ratio decreases, approaching the typical OC/EC ratio of vehicular emissions."

2. Figure 2a: Please clarify the term "corrected PM2.5." What does this correction entail?

**Response:** The reasons to correct SHARP PM$_{2.5}$ mass concentrations are included in the updated manuscript and supplement. Please refer to our reply in response to comment 3 raised by reviewer #1.

3. Figure S3: The error bar (uncertainty) in Figure S3a should be defined for clarity. Additionally, in Figure S3b, the legend should be corrected to read "SOC_u/SOC_c".

**Response:** Revised as suggested. Please find the revised Figure S3 (new Figure S5 in the revised manuscript).

[Figure]

**Figure R1 (New Figure S5).** Comparison of (a) absolute concentration and (absolute uncertainties represented in error bar), and (b) relative uncertainty of POC and SOC between BI with sulfate and ammonium as tracers for SOC.

4. Line 290: Based on Figure 3, it appears that SOC values are slightly higher on weekends, particularly in winter and spring. Could you provide any insights into this observation?

**Response:** It is well documented that secondary organic aerosol formation is influenced by various factors, such as the organic precursors and atmospheric oxidants levels, as well as meteorological conditions. As shown in Figure 3, SOC values are slightly higher on weekends, which is similar to O$_3$ patterns. The slightly higher O$_3$ levels on weekends levels than weekdays could be due to the weak titration effects due to the reduced NO$_x$ from vehicle emissions or other anthropogenic emissions during weekends. Thus, the slightly higher SOC levels on weekends would be attributed to the stronger atmospheric oxidation capacity.

The following sentence will be added to describe the difference of SOC variations on weekdays and weekend.

"The weekday-weekend patterns of POC and SOC displayed notable distinctions. Specifically, SOC was slightly higher on weekends, whereas enhancement of POC was found on weekdays throughout different seasons. Similar higher weekend levels were found for O$_3$, which could be due to the weak titration effects due

to the reduced NO$_x$ from vehicle emission or other anthropogenic emissions during weekends. This observation suggests that anthropogenic emissions exerted a stronger influence on POC levels, while SOC levels appeared to be more influenced by the active photochemistry VOCs emissions from the nearby broadleaf woods rather than the anthropogenic sources."

5. Line 298: The reference to daily ozone patterns appears to be incorrectly cited as Figure S4b; it should be Figure 3b.

**Response:** Revised as suggested.

6. Line 372: The term "ensuring analysis" seems unclear to me. Could you provide a more detailed explanation or rephrase it for clarity?

**Response:** To clarify, the sentence has been revised as follows and will be incorporated in the revised manuscript.

"In this work, PM$_{2.5}$ episodes were identified as periods of hourly concentrations exceeding 25 μg m$^{-3}$ and lasting 6 consecutive hours or longer at more than three monitoring stations."

7. Figure 6: For the non-episode data, is the representation limited to the hours at the start of each season, or does it encompass all non-episode hours throughout the respective season?

**Response:** The non-episode data in Figure 6 is the combination of all non-episode hours in individual season. The caption of Figure 6 is revised to clarify the statement. The revised caption is copied here for easy reference.

"Figure 6. Comparison of select pollutant levels during episodes and non-episodes for individual episodes. The comparison parameters include concentrations of (a) O$_3$, (b) NO$_x$, (c) PM$_{2.5}$, (d) POC, and (e) SOC, (f) POC and SOC percentage contributions, and mass increment ratios of (g) O$_3$ and NO$_x$, (h) PM$_{2.5}$, and (i) POC and SOC. In panels (a)-(e), the filled squares represent during-episode concentrations while the empty circles represent the combination of all non-episode hours throughout the individual season. In panels (g)-(i), the light-yellow shaded zone marks the mass increment ratios (calculated as mass concentration during the episode divided by that during the non-episode hours in the same season) values of less than 1.

---

## Author Response (AR2)

**Response to Review Comments by Editor on "Bayesian Inference-Based Estimation of Hourly Primary and Secondary Organic Carbon at Suburban Hong Kong: Multi-temporal Scale Variations and Evolution Characteristics during PM$_{2.5}$ episodes" by S. Wang et al.**

We thank the editor for the detailed comments. Below is our point-by-point response to each comment, marked in blue. The revised text in the main manuscript is also marked in blue.

1. I suggest to delete the statement of "with sulfate identified as the most suitable SOC tracer" in Line 20 to avoid misleading of introducing sulfate as a "recommended" tracer.

**Response:** Suggestion taken. The statement has been deleted in the revised manuscript.

2. In Line 250-251: It is better to provide more specific discussion about where and when SOC formation pathways could be different from sulfate from our current understanding. For example, southeastern US, NCP in China, and the Amazon forest are all interested area where the SOA formation and sulfate origin are distinct.

**Response:** We thank the reviewer for this suggestion and made the following changes.

Lines 248-258:

> Considering the similarities in formation pathways, the BI-SO$_4^{2-}$ model would yield more accurate estimations when regional transport has a stronger influence compared to local formation processes. Conversely, when SOC formation pathways are significantly disconnected in time and in space from those of sulfate, the performance of the BI-SO$_4^{2-}$ model would be less satisfactory. For example, in clean regions like the southeast US and Amazon where SOA were dominated by fast local oxidation chemistry of biogenic VOCs (Xu et al., 2015; Riemer et al., 1998; Langford et al., 2022), sulfate may not serve as a good tracer to track SOA in the BI-SO$_4^{2-}$ model. In urban areas where daytime photochemical processing may play a significant role in SOA formation, e.g., summertime Beijing (Duan et al., 2020), sulfate may also fail as a proper tracer. Thus, an integrative evaluation of available PM composition, along with related air pollutant and meteorological conditions, is recommended to aid identification of a suitable SOC tracer in implementing the BI method, as well as assessing the interpretability of the BI method derived POC and SOC data.

References:

Duan, J., Huang, R.-J., Li, Y., Chen, Q., Zheng, Y., Chen, Y., Lin, C., Ni, H., Wang, M., Ovadnevaite, J., Ceburnis, D., Chen, C., Worsnop, D. R., Hoffmann, T., O'Dowd, C., and Cao, J.: Summertime and wintertime atmospheric processes of secondary aerosol in Beijing, Atmos. Chem. Phys., 20, 3793-3807, http://doi.org/10.5194/acp-20-3793-2020, 2020.

Langford, B., House, E., Valach, A., Hewitt, C. N., Artaxo, P., Barkley, M. P., Brito, J., Carnell, E., Davison, B., MacKenzie, A. R., Marais, E. A., Newland, M. J., Rickard, A. R., Shaw, M. D., Yáñez-Serrano, A. M., and Nemitz, E.: Seasonality of isoprene emissions and oxidation products above the remote Amazon, Environmental Science: Atmospheres, 2, 230-240, http://doi.org/10.1039/d1ea00057h, 2022.

Riemer, D., Pos, W., Milne, P., Farmer, C., Zika, R., Apel, E., Olszyna, K., Kliendienst, T., Lonneman, W., Bertman, S., Shepson, P., and Starn, T.: Observations of nonmethane hydrocarbons and oxygenated volatile organic compounds at a rural site in the southeastern United States, J. Geophys. Res.-Atmos., 103, 28111-28128, http://doi.org/10.1029/98jd02677, 1998.

Xu, L., Guo, H., Boyd, C. M., Klein, M., Bougiatioti, A., Cerully, K. M., Hite, J. R., Isaacman-VanWertz, G., Kreisberg, N. M., Knote, C., Olson, K., Koss, A., Goldstein, A. H., Hering, S. V., de Gouw, J., Baumann, K., Lee, S. H., Nenes, A., Weber, R. J., and Ng, N. L.: Effects of anthropogenic emissions on aerosol formation from isoprene and monoterpenes in the southeastern United States, Proc Natl Acad Sci U S A, 112, 37-42, http://doi.org/10.1073/pnas.1417609112, 2015.

3. The authors highlight the potential nighttime aqueous-phase reactions for SOC. Please clarify whether this analysis would be affected by the choice of sulfate as the SOC tracer?

**Response:** Previous field measurements and chamber studies have observed significant SOA (or SOC) enhancement with increasing water uptake and acidification, highlighting the important role of aqueous phase reactions in facilitating secondary organic aerosol formation (Huang et al., 2018; Mcneill et al., 2012; Guo et al., 2012). Recent studies pointed out that abundant aerosol water content and acidic conditions may also notably promote secondary sulfate formations in ambient environment (Huang et al., 2023; Wang et al., 2019). Furthermore, it has been reported that SOA could be directly mediated by the abundance of sulfate (Xu et al., 2015).

In this study, we observed positive correlations between BI-$SO_4^{2-}$ SOC and AWC and acidity during nighttime episodic hours, suggesting potential nocturnal aqueous-phase reactions for SOC formation. We further explore the relationships of SOC with AWC and acidity by using $NH_4^+$ as SOC tracer. As shown in Figure R1, the relationships are less significant compared with those using $SO_4^{2-}$. These results are reasonable since the BI-$NH_4^+$ SOC have higher uncertainties with less satisfied model performance (indicating by BIC) compared with BI-$SO_4^{2-}$ SOC. Due to the challenge in directly measuring AWC and acidity, we use ISOPPIRA II model to compute the AWC and acidity, in which sulfate is an important model input. Therefore, the effects of sulfate on influencing the AWC and acidity simulation, and SOC results could not be ruled out. Sulfate, as an abundant secondary constituent of $PM_{2.5}$, is a key player in multiple formation pathways of SOA. There are chemical and physical basis for our conclusion of the importance of potential nighttime aqueous-phase reactions for SOC. We do not think it is an "artifact" of selecting sulfate as the SOC tracer in our BI-method.

[Figure]

**Figure R1**. Scatter plot of SOC with (a) aerosol water contents (AWC) and (b) [H$^+$] during the winter haze episodes. The solid red circle represents daytime hours, blank green circle represents nighttime hours.

References:

Guo, S., Hu, M., Guo, Q., Zhang, X., Zheng, M., Zheng, J., Chang, C. C., Schauer, J. J., and Zhang, R.: Primary sources and secondary formation of organic aerosols in Beijing, China, Environ. Sci. Technol., 46, 9846-9853, http://doi.org/10.1021/es2042564, 2012.

Huang, D. D., Zhang, Q., Cheung, H. H. Y., Yu, L., Zhou, S., Anastasio, C., Smith, J. D., and Chan, C. K.: Formation and Evolution of aqSOA from Aqueous-Phase Reactions of Phenolic Carbonyls: Comparison between Ammonium Sulfate and Ammonium Nitrate Solutions, Environ Sci Technol, 52, 9215-9224, http://doi.org/10.1021/acs.est.8b03441, 2018.

Huang, X., Liu, Z., Ge, Y., Li, Q., Wang, X., Fu, H., Zhu, J., Zhou, B., Wang, L., George, C., Wang, Y., Wang, X., Su, J., Xue, L., Yu, S., Mellouki, A., and Chen, J.: Aerosol high water contents favor sulfate and secondary organic aerosol formation from fossil fuel combustion emissions, Npj Clim Atmos Sci, 6, http://doi.org/10.1038/s41612-023-00504-1, 2023.

McNeill, V. F., Woo, J. L., Kim, D. D., Schwier, A. N., Wannell, N. J., Sumner, A. J., and Barakat, J. M.: Aqueous-phase secondary organic aerosol and organosulfate formation in atmospheric aerosols: a modeling study, Environ Sci Technol, 46, 8075-8081, http://doi.org/10.1021/es3002986, 2012.

Wang, H., Ding, J., Xu, J., Wen, J., Han, J., Wang, K., Shi, G., Feng, Y., Ivey, C. E., Wang, Y., Nenes, A., Zhao, Q., and Russell, A. G.: Aerosols in an arid environment: The role of aerosol water content, particulate acidity, precursors, and relative humidity on secondary inorganic aerosols, Sci Total Environ, 646, 564-572, http://doi.org/10.1016/j.scitotenv.2018.07.321, 2019.

Xu, L., Guo, H., Boyd, C. M., Klein, M., Bougiatioti, A., Cerully, K. M., Hite, J. R., Isaacman-VanWertz, G., Kreisberg, N. M., Knote, C., Olson, K., Koss, A., Goldstein, A. H., Hering, S. V., de Gouw, J., Baumann, K., Lee, S. H., Nenes, A., Weber, R. J., and Ng, N. L.: Effects of anthropogenic emissions on aerosol formation from isoprene and monoterpenes in the southeastern United States, Proc Natl Acad Sci U S A, 112, 37-42, http://doi.org/10.1073/pnas.1417609112, 2015.

4. Please provide clear descriptions for figures.

Figure 1a and 2a: The error bars are not explained.

**Response:** Suggestion taken.

The caption has been revised as "Figure 1. (a) Box plot of $\overline{K_1}$ and $\overline{K_2}$ values across different seasons (the squares and horizontal lines in the box denote the average and median, the lower and upper boundaries of the boxes represent the $25^{th}$ and $75^{th}$ percentile values, and whisker are $10^{th}$ and $90^{th}$ percentile). (b) The diurnal variations of $\overline{K_1}$ and $\overline{K_2}$ in individual seasons (solid lines represent the average values, area indicate one standard deviation of the results)."

The caption has been revised as "Figure 2. (a) Time series of meteorological parameters (wind speed, wind direction, temperature, and RH), gaseous pollutants ($O_3$, and $NO_x$), $PM_{2.5}$ (the red dash line marks the WHO AQG IT-4 value), OC and EC, as well as POC and SOC (the $y$-axis error bars represent uncertainties derived from BI method) and (b) Seasonal variations (the circle and horizontal lines in the box denote the average and median, the lower and upper boundaries of the boxes represent the $25^{th}$ and $75^{th}$ percentile values, and whisker are $10^{th}$ and $90^{th}$ percentile) during the observation period (16 July 2020–31 December 2021) at the HKUST supersite."

Figure 2b: Box-whistler information is missing.

**Response:** Suggestion taken. The box-whisker has been defined in the caption.

Figures 4 and 5: Aren't the box bottom and top 25th and 75th, respectively? What are exactly the whistlers?

**Response:** The box-whisker information has been included in the caption.

"Figure 4. Concentrations of SOC as a function of (a) temperature bins and (b) RH bins under different $PM_{2.5}$ groups in individual seasons during the entire measurement period (the circles and horizontal lines in the box denote the average and median, the lower and upper boundaries of the boxes represent the $25^{th}$ and $75^{th}$ percentile values, and whisker are $10^{th}$ and $90^{th}$ percentile).

"Figure 5. Concentrations of SOC as a function of (a) $O_x$ bins (b) $NO_x$ bins under different $PM_{2.5}$ groups in individual seasons during the entire measurement period (the circles and horizontal lines in the box denote the average and median, the lower and upper boundaries of the boxes represent the $25^{th}$ and $75^{th}$ percentile values, and whisker are $10^{th}$ and $90^{th}$ percentile).

Figure 6a: What are the error bars?

**Response:** The error bars have been defined in the revised manuscript. They are copied here for easy reference:

"Figure 6. Comparison of select pollutant levels during episodes and non-episodes for individual episodes. The comparison parameters include concentrations of (a) $O_3$, (b) $NO_x$, (c) $PM_{2.5}$, (d) POC, and (e) SOC, (f) POC and SOC percentage contributions, and mass increment ratios of (g) $O_3$ and $NO_x$, (h) $PM_{2.5}$, and (i) POC and SOC. In panels (a)-(e), the filled squares represent the average values during-episode concentrations while the empty circles represent the average of all non-episode hours throughout the individual season, the error bars represent one standard deviation of the results. In panels (g)-(i), the light-yellow shaded zone marks the mass increment ratio (calculated as mass concentration during the episode divided by that during the non-episode hours in the same season) values of less than 1.

Figure 7b and 8b are too small. The box and whistler information is missing. The stars are not explained.

**Response:** Suggestion taken. Figure 7b and 8b have been modified to be clearer.

The caption has been revised as

"Figure 7. SOC variation characteristics during typhoon episodes in summer 2021. (a) Time series of meteorological parameters (wind speed and direction), gaseous pollutants ($O_3$ and $NO_x$), $PM_{2.5}$ mass concentrations, POC and SOC levels and their relative percentage contributions, with the yellow shadow area marking individual episode periods of EP45-51. (b) Concentrations of SOC as a function of temperature, RH, $O_x$ ($O_3+NO_2$) and $NO_x$ bins, with daytime and nighttime episode hours plotted separately (the squares and horizontal lines in the box denote the average and median, the lower and upper boundaries of the boxes represent the $25^{th}$ and $75^{th}$ percentile values, and whisker are $10^{th}$ and $90^{th}$ percentile. Significance level (p) by t-test: ****$p < 0.0001$, ***$0.0001 < p < 0.001$, **$0.001 < p < 0.01$, *$0.01 < p < 0.05$).

"Figure 8. SOC variation characteristics during haze episodes in winter 2020 and 2021. (a) Time series of meteorological parameters (wind speed and direction), gaseous pollutants ($O_3$ and $NO_x$), $PM_{2.5}$ mass concentrations, POC and SOC levels and their relative percentage contributions, with the yellow shadow area marking individual episodes (EP10-13 and EP62-65). (b) Concentrations of SOC as a function of temperature, RH, $O_x$ ($O_3+NO_2$) and $NO_x$ bins, with daytime and nighttime episode hours plotted separately (the squares and horizontal lines in the box denote the average and median, the lower and upper boundaries of the boxes represent the $25^{th}$ and $75^{th}$ percentile values, and whisker are $10^{th}$ and $90^{th}$ percentile. Significance level (p) by t-test: ****$p < 0.0001$, ***$0.0001 < p < 0.001$, **$0.001 < p < 0.01$, *$0.01 < p < 0.05$).